# Dopey1-Mon2 complex binds to dual-lipids and recruits kinesin-1 for membrane trafficking

Divyanshu Mahajan[1], Hieng Chiong Tie [1], Bing Chen[1] & Lei Lu [1]

Proteins are transported among eukaryotic organelles along the cytoskeleton in membrane carriers. The mechanism regarding the motility of carriers and the positioning of organelles is a fundamental question in cell biology that remains incompletely understood. Here, we find that Dopey1 and Mon2 assemble into a complex and localize to the Golgi, endolysosome and endoplasmic reticulum exit site. The Golgi localization of Dopey1 and Mon2 requires their binding to phosphatidylinositol-4-phosphate and phosphatidic acid, respectively, two lipids known for the biogenesis of membrane carriers and the specification of organelle identities. The N-terminus of Dopey1 further interacts with kinesin-1, a plus-end or centrifugal-direction microtubule motor. Dopey1-Mon2 complex functions as a dual-lipid-regulated cargo-adaptor to recruit kinesin-1 to secretory and endocytic organelles or membrane carriers for centrifugally biased bidirectional transport. Dopey1-Mon2 complex therefore provides an important missing link to coordinate the budding of a membrane carrier and subsequent bidirectional transport along the microtubule.

[1] School of Biological Sciences, Nanyang Technological University, 60 Nanyang Drive, Singapore 637551, Singapore. Correspondence and requests for materials should be addressed to L.L. (email: lulei@ntu.edu.sg)

Proteins and lipids (cargos) are dynamically exchanged among eukaryotic organelles in membrane carriers. The implementation of organelle positioning and carrier trafficking requires the dynein-mediated centripetal (minus-end directed) and kinesin-mediated centrifugal (plus-end directed) movement along microtubules[1,2]. Despite the significant advancement made in this field, there still lacks a clear molecular understanding of this fundamental and ubiquitous process[3]. For example, kinesin-1, the most ubiquitous plus-end directed motor, has long been known to play important roles in membrane trafficking and organelle positioning. However, how kinesin-1 is molecularly coupled to membrane compartments is still unknown in most cases. It has been established that the biogenesis of membrane carriers from the donor compartment can be assisted by lipids and microtubule-based motors[4]. In this process, lipids facilitate the generation of membrane curvature and recruit diverse effectors for coat formation and cargo selection. Microtubule-based motors enforce the bending, tubulation, and scission of the donor membrane to aid in the production of carriers. It is nonetheless unclear how carrier biogenesis and the subsequent motility along the microtubule tracks are coordinated.

Dopey1 and Mon2 are poorly characterized peripheral Golgi membrane proteins. The primary sequence of Mon2 is related to Sec7-family guanine nucleotide exchange factors for Arf GTPases, although we previously demonstrated that it does not possess the guanine nucleotide exchange activity[5]. In the current study, we discover that Dopey1 and Mon2 assemble into a complex and further elucidate their cellular function as a phosphatidic acid (PA) and phosphatidylinositol-4-phoshate (PI4P)-dependent kinesin-1 adaptor to transport membrane carriers or position organelles along the microtubule network.

## Results

**Mon2 homodimer interacts with two monomeric Dopey1**. To identify interacting partners of Mon2, we performed a large-scale immunoprecipitation (IP) using Mon2 antibody followed by mass-spectrometry. Dopey1 was found to be one of the top hits. It became our focus since yeast Mon2p has been reported to interact with Dopey1 ortholog, Dop1p[6]. There are two Dopey orthologs in mammals, Dopey1 and 2, which share an identity of 36%. We raised antibodies that can specifically recognize each (Supplementary Fig. 1a–e). The specific interaction between native Dopey1 and Mon2 was confirmed by reciprocal co-IPs (Fig. 1a). Unexpectedly, Dopey2 did not interact with Mon2, demonstrating that it might have distinct cellular functions from Dopey1.

Similarly, exogenously transfected Dopey1 and Mon2 were able to co-IP each other (Fig. 1b; Supplementary Fig. 1f, g). Serial truncation of Mon2 revealed that the C-terminal region of ~260 residues (1458–1718), which is referred to as Mon2 extreme C-terminal region (MEC), was necessary and sufficient for its interaction with Dopey1 (Fig. 1b). The central region of Dopey1 (residues 540–1900) (Fig. 1c), but not its N- or C-terminus (Fig. 1d), was found to interact with Mon2. Since expression levels of transfected Dopey1 and Mon2 far exceeded those of corresponding endogenous ones (Supplementary Fig. 1h–k), their interactions might be direct. Using in vitro translated proteins, we found that MEC co-IPed the Dopey1 fragment encompassing residues 894–1247 (Fig. 1e), demonstrating the direct interaction between Mon2 and Dopey1.

We discovered that GFP-Mon2 was able to co-IP DMyc-Mon2 (Fig. 1f). Under the total internal reflection fluorescence (TIRF) microscopy, when cell-expressed GFP-Mon2 was immobilized on the surface of a glass coverslip, 90% GFP-fluorescence objects displayed two photo-bleaching steps (top of Fig. 1g; supplementary Fig. 1l, m), implying that Mon2 dimerizes. The dimerization

should be mediated by MEC since it interacted with both itself (cell-expressed or in vitro translated) and full length Mon2 (Fig. 1f, h) and an intact MEC is required for Mon2's self-interaction (Supplementary Fig. 1n). In contrast to Mon2, full length Dopey1 is likely a monomer since two differently tagged Dopey1 did not co-IP each other when co-expressed (Fig. 1i) and 95% GFP-Dopey1 positive objects had single photo-bleaching steps (middle of Fig. 1g). Interestingly, the self-interaction of overexpressed Dopey1 became detectable in the presence of co-expressed Mon2 or MEC (Fig. 1i); furthermore, co-expression of DMyc-Mon2 greatly increased the fraction of GFP-Dopey1 positive objects with two photo-bleaching steps from 5 to 68% (middle and bottom of Fig. 1g). Collectively, our data suggest that Mon2 homodimer interacts with two copies of Dopey1 to form a heterotetrameric complex.

**The Golgi localization of endogenous Dopey1 depends on Mon2**. Similar to yeast Dop1p[6], mammalian Dopey1 and 2 localized to the Golgi (Fig. 2a). Like Mon2[5], Dopey1 and 2's Golgi localization is sensitive to brefeldin A (Supplementary Fig. 2a, b), an Arf1 inhibitor[7]. Since Dopey1 and Mon2 can form a protein complex, we investigated the role of one in the Golgi targeting of the other. To that end, we developed siRNAs to specifically knockdown Mon2, Dopey1, or Dopey2, and found that their cellular stabilities are independent of each other (Supplementary Fig. 2c, d). While the depletion of Dopey1 did not affect the Golgi localization of Mon2, the depletion of Mon2 caused a great loss of Dopey1 at the Golgi (Fig. 2b–e), which was rescued by the expression of an RNAi-resistant Mon2 (Supplementary Fig. 2e). In contrast, the Golgi localization of Dopey2 was not affected by Mon2 depletion (Supplementary Fig. 2f). Hence, our observation demonstrated that the Golgi localization of endogenous Dopey1 requires Mon2 but not vice versa and is consistent with a previous finding made in the yeast[6].

**DEC is responsible for the Golgi targeting of Dopey1**. Serial truncation analysis of Dopey1 was employed to investigate the domain/region that is responsible for its Golgi targeting (Fig. 2f, g and Supplementary Fig. 2g). A C-terminal fragment comprising residues 2146–2456 was found to be the smallest fragment that displayed the Golgi localization (Fig. 2f, g). The corresponding region of this fragment is subsequently referred to as Dopey1 extreme C-terminal region (DEC). Deletion of DEC made Dopey1 lose its Golgi localization (Fig. 2f, g). By examining its primary sequence, we identified a highly conserved LKRL motif at residues 2169–2172 (Fig. 2h). When the motif is mutated to AAAA (LKRL-4A), both DEC and full length Dopey1 lost their Golgi localization and became cytosolic (Fig. 2f, g). Together, our data demonstrate that DEC is both necessary and sufficient for the Golgi targeting of Dopey1. In contrast, corresponding fragments of Dopey2 were all cytosolic (Supplementary Fig. 2h, i) despite sharing a high sequence similarity (Fig. 2h), implying that Dopey1 and 2 have distinct Golgi localization mechanism.

Although GFP-Dopey1ΔDEC lost its Golgi localization (Supplementary Fig. 2j), it is expected (Fig. 1c, e) and later confirmed to interact with Mon2 (see below). Interestingly, we found that overexpressed DMyc-Mon2 recruited a significant amount of GFP-Dopey1ΔDEC to the Golgi (Supplementary Fig. 2j). When GFP-Dopey1ΔDEC was artificially tethered to the peroxisome (see below), endogenous Mon2 or exogenously expressed DMyc-MEC also localized to the peroxisome (Supplementary Fig. 2k, l). Though these experiments might not reflect native scenario, their reciprocal membrane recruitments upon overexpression further confirmed the interaction between Dopey1 and Mon2.

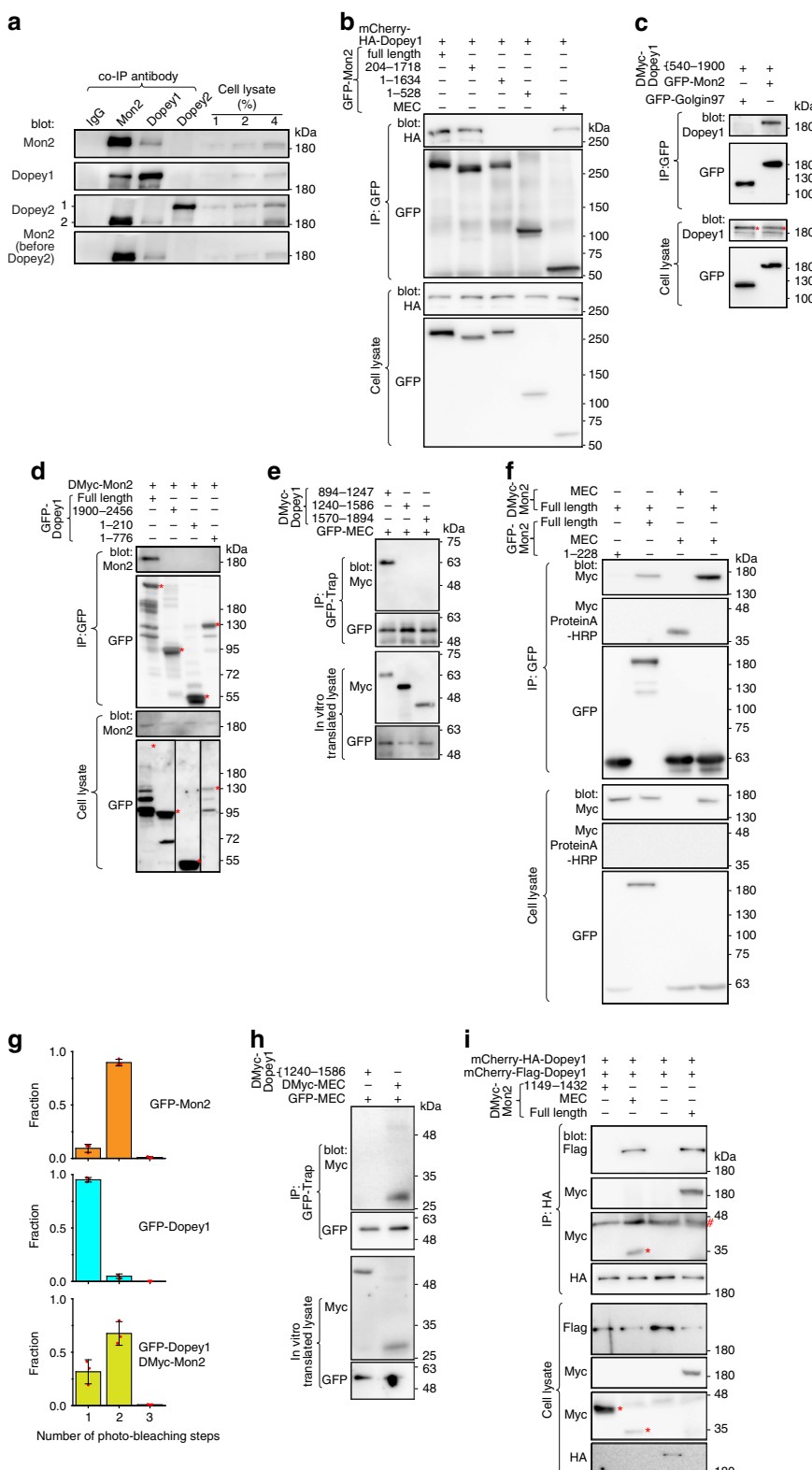

**DEC binds to PI4P**. DEC is neither necessary (Fig. 1c and Supplementary Fig. 3a) nor sufficient (Fig.1d) for Dopey1–Mon2 interaction. Therefore, its Golgi targeting must depend on its association with an unidentified Golgi factor. It is difficult to know the identity of DEC-interacting factor without systematic screening of the Golgi proteome and lipidome. However, a few lipid species are known to be specifically enriched on the cytosolic leaflet of the Golgi membrane, including PI4P, PA, lyso-PA

(LPA), and diacylglycerol (DAG)[8–10]. We took advantage of small molecule inhibitors to study the effect of these candidate lipids on the Golgi localization of DEC. It was found that, within 10 min treatment of phenylarsine oxide (PAO), an inhibitor of cellular PI4P production (Supplementary Fig. 3b)[11,12], tandem-dimer DEC (tdDEC) (Fig. 3a, b), GFP-tagged (Supplementary Fig. 3c, d) and endogenous Dopey1 (Fig. 3c, d) lost their Golgi localization. The dissociation of Dopey1 from the Golgi was not due to the

**Fig. 1** Mon2 homodimer interacts with two Dopey1 monomers. **a** Endogenous Dopey1 and Mon2 interact with each other. Cell lysates were subjected to IP followed by immunoblotting using indicated antibodies. The first and third row are from the same protein blot sequentially probed by Mon2 and Dopey2 antibodies. Before Dopey2 antibody blotting, the corresponding region of the third image is shown in the last row. 1 and 2 indicate Dopey2 and Mon2 respectively. **b** MEC interacts with Dopey1. Cell lysates expressing indicated tagged proteins were subjected to IP followed by immunoblotting using indicated antibodies. **c** The central region of Dopey1 interacts with Mon2. **d** The N and C-terminal regions of Dopey1 do not interact with Mon2. Strips were cropped from the same blot image and rearranged in the last row. **e** MEC directly interacts with the central region of Dopey1. Indicated proteins were mixed after in vitro translation and the mixture was subjected to IP using GFP-Trap agarose beads. **f** Mon2 interacts with itself via MEC. **g** The stoichiometry of GFP-fusion protein complexes by counting single-molecule photo-bleaching steps. Indicated GFP-fusion proteins expressed in cell lysates were immobilized onto glass coverslips and subjected to time lapse photo-bleaching under TIRF microscopy. The histogram of analyzable objects are shown. Red dot, individual data point; error bar, mean ± s.d. from $n = 3$ independent experiments. **h** MEC directly interacts with itself. The experiment was similar to **e**. **i** Dopey1 interacts with itself only in the presence of Mon2 or MEC. IPs in **c**–**f** and **h**, **i** were similarly performed as those in (**b**). In **c**, **d** and **i**, * and # indicate specific and nonspecific band, respectively. Molecular weights (in kDa) are labeled in all gel blots. All cell lysates were from HEK293T cells. Source data are provided as a Source Data file

dissociation of Mon2, which remained unchanged even after prolonged PAO treatment (Fig. 3e, f). In contrast, the Golgi localization of endogenous Dopey2 was insensitive to PAO (Supplementary Fig. 3e, f). Further supporting our PAO experiment, when the major Golgi phosphatidylinositol 4-kinase, PI4KIIIβ[13], was depleted by RNAi (Supplementary Fig. 3g, h), we observed that the Golgi localization of Dopey1, but not Mon2, also decreased (Fig. 3g, h).

The sensitivity of both Dopey1 and DEC to PI4P-inhibitor suggests a possible direct binding to PI4P. Indeed, we observed that significantly more GFP-tdDEC was pulled down by PI4P than PI, PI3P, PI5P, and PI(4,5)P$_2$ liposomes (Fig. 3i). As controls, SNX3 and P4M displayed positive interactions with PI3P and PI4P, respectively, as expected (Fig. 3i, j)[14,15]. Overexpressed full length Dopey1 and DEC-containing fragment (1900–2456), but not the one with LKRL-4A mutation, were also robustly pulled down by PI4P-liposome (Fig. 3k, l). The interaction between DEC and PI4P should be direct as PI4P-liposome specifically bound purified GST-DEC, but not the one harboring LKRL-4A mutation (Fig. 3m). Using a series of diluted GST-DEC in PI4P-liposome-mediated pull-downs, we estimated the apparent affinity of DEC–PI4P interaction to be ~1 μM (Supplementary Fig. 3i, j). The affinity is in the same range as PI4P-binding protein GOLPH3/Vps74p[16], but much weaker than that of P4M[17], which was measured to be ~100 nM in our assay (Supplementary Fig. 3k, l). The weaker affinity of DEC is consistent with its more sensitive response to PAO (Supplementary Fig. 3m–o).

In summary, the Golgi targeting of Dopey1 seems to require the specific binding of DEC to PI4P, the phosphoinositide characteristic of the Golgi[10,18,19]. Hence, our data uncovered DEC as a PI4P-binding module.

**Dimerized Dopey1 targets to the Golgi independent of Mon2.** Since Dopey1 alone can bind to PI4P, its dependence on Mon2 for Golgi localization seems puzzling. We first ruled out the possibility that Mon2 is essential for the biogenesis of Golgi PI4P (Supplementary Fig. 3p–s). Next, we noticed that single DEC probably had a much weaker PI4P binding affinity than tdDEC as evidenced by its much faster Golgi dissociation rate during PAO treatment (Fig. 3a, b and Supplementary Fig. 3m, n). In fact, GST-DEC protein used in our binding assays should be a dimer due to homodimerization by GST. To further investigate the effect of dimerization on Dopey1's Golgi targeting, we constructed a chimera by replacing the Mon2-interacting central region (residue 540–1900) of Dopey1 with ACC1, an artificial dimerization domain[20] (Supplementary Fig. 3t). In contrast to GFP-Dopey1 (Fig. 3n), the Golgi localization of Dopey1-ACC1 was no longer sensitive to Mon2 depletion (Fig. 3o). Our observation is consistent with the conventional notion that

homodimerization provides bivalent binding to a ligand, especially for phosphoinositide-binding domains[21]. Monovalent PI4P interaction alone might be insufficient to stably anchor Dopey1 onto the membrane. Therefore, Mon2 is probably required to homodimerize Dopey1 for the latter's bivalent engagement with PI4P.

**The Golgi localization of Mon2 requires PA.** We previously demonstrated that the Golgi localization of Mon2 requires its intact N-terminus[5]. Our further investigation revealed that a fragment encompassing residues 1–1200 showed a weak but significant Golgi localization in live cell imaging (Fig. 4a, b; Supplementary Fig. 4a). Further truncation of this clone abolished its Golgi localization. Similar to Dopey1, we asked how Mon2 is targeted to the Golgi. We discovered that R59949, a diacylglycerol kinase inhibitor which can reduce cellular PA[22–24], dissociated Mon2 from the Golgi (Fig. 4c, d). Endogenous Dopey1 (Supplementary Fig. 4b, c), but not Dopey1-ACC1 (Supplementary Fig. 4d, e), also dissociated from the Golgi under this treatment, as expected from its dependence on Mon2 for the Golgi localization.

We subsequently tested the possible interaction between Mon2 and PA. It was found that exogenously expressed Mon2 and Mon2(1–1200) were selectively pulled down by PA-liposome, but not phosphatidylethanolamine (PE), phosphatidylcholine (PC), and phosphatidylserine (PS) liposomes and liposomes containing PI, LPA, DAG, or PI4P (Fig. 4e), which can be metabolically synthesized from PA[25]. Further truncation of Mon2(1–1035) abolished its PA-binding (Fig. 4f), suggesting that the fragment comprising residue 1–1035, including conserved DCB, HUS, and HDS1–2 regions, is the smallest region that either directly or indirectly interacts with PA. However, Mon2(1–1035) did not localize to the Golgi in vivo (Fig. 4a, b), in contrast to 1–1200, demonstrating that PA-binding is required but is probably insufficient for the Golgi localization of Mon2. Resolving direct or indirect PA-binding and identifying additional factor(s) for Mon2's Golgi localization await further investigation. In summary, our data so far demonstrate that Dopey1–Mon2 complex probably engages two types of Golgi resident lipids, PA and PI4P, for its Golgi targeting.

**Dopey1 interacts with kinesin-1 through its N-terminus.** To investigate the cellular function of Dopey1, a large-scale IP followed by mass-spectrometry was employed to explore potential binding partners of Dopey1. Besides Mon2, Kif5b, the heavy chain of kinesin-1, was identified as one of the top hits. The interaction between Dopey1 and Kif5b was confirmed by subsequent endogenous co-IPs. We found that, in addition to Mon2, Dopey1, but not Dopey2, specifically co-IPed Kif5b, but not

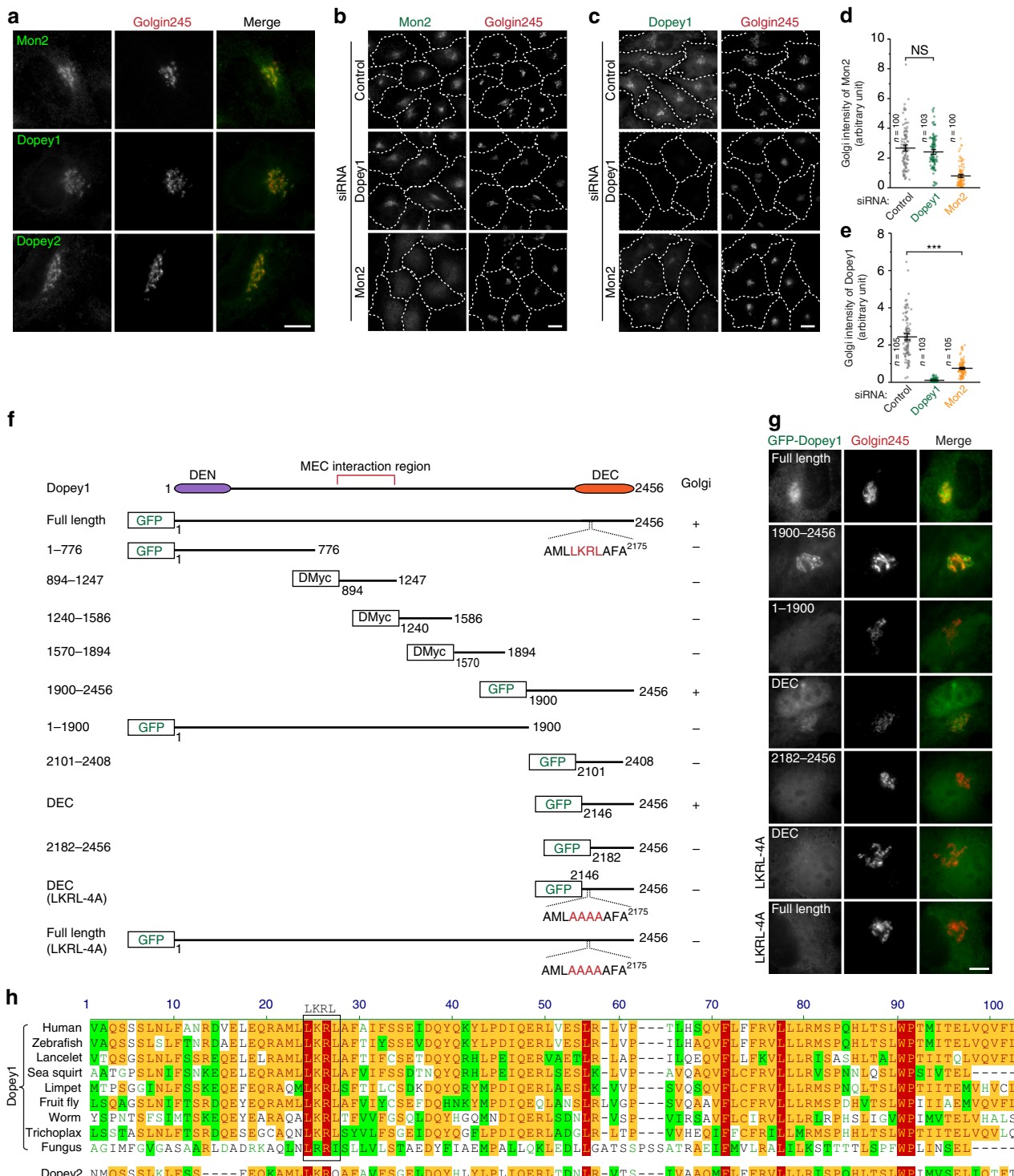

**Fig. 2** The Golgi localization of Dopey1. **a** Endogenous Dopey1 and Mon2 localize to the Golgi. Cells were subjected to immunofluorescence to co-stain Mon2, Dopey1, or Dopey2 together with Golgin245 (a Golgi marker). **b–e** The Golgi localization of endogenous Dopey1 requires Mon2. Cells were subjected to knockdown by indicated siRNAs followed by immunofluorescence to reveal indicated proteins in (**b**, **c**). Cell contours are marked by white dotted lines. The Golgi fluorescence intensity of Mon2 (**d**) and Dopey1 (**e**) per cell was subsequently quantified. *n*, the number of cells analyzed; error bar, mean ± s.e.m. *P* values are from *t* test (unpaired and two-tailed). N.S., not significant; ***, $P \leq 0.0005$. **f**, **g** DEC is necessary and sufficient for the Golgi localization of Dopey1. Schematic diagrams of Dopey1 truncations and mutants are shown in (**f**). The Golgi localization is scored based on fluorescence staining experiment in (**g**), in which cells expressing selected Dopey1 truncations and mutants were stained for Golgin245. **h** Multiple alignment of metazoan protein sequences of DEC. Uniprot IDs used are B2RWN9 (human), F1R226 (zebrafish), C3ZLF8 (lancelet), F7BCQ8 (sea squirt), V3ZCY9 (limpet), A1ZBE8 (fruit fly), Q9XW10 (worm), B3S9K0 (trichoplax), and Q9Y7B3 (fungus). The corresponding region of human Dopey2 (Uniprot ID: Q9Y3R5) is also included below. Scale bar, 10 μm. All cells are HeLa cells. Source data are provided as a Source Data file

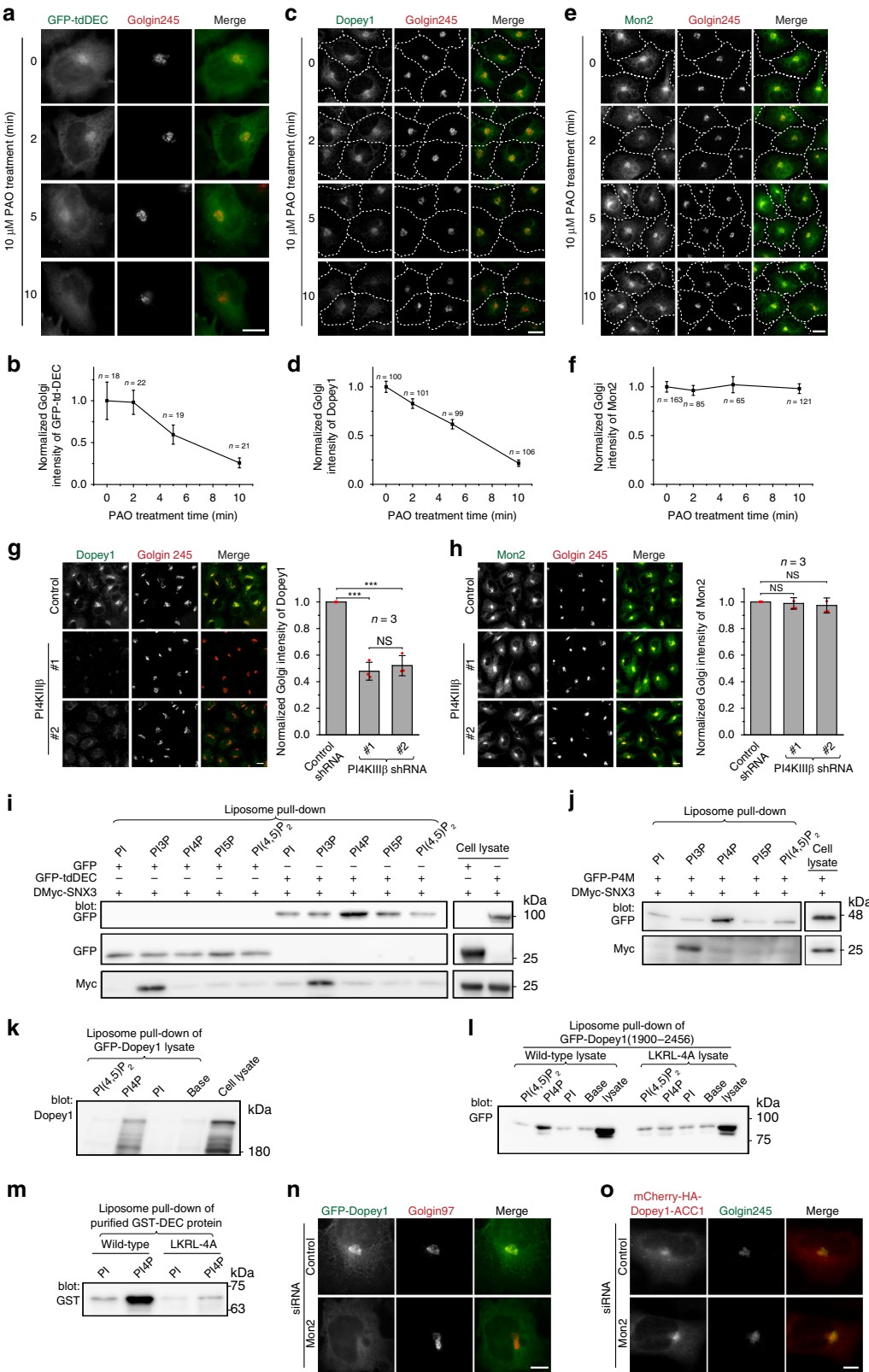

Kif20a and Kif18a (Fig. 5a). Kinesin-1 is a heterotetramer consisting of a kinesin heavy chain (Kif5) and kinesin light chain (KLC) homodimer[1,26]. The most ubiquitous and predominant isoforms of kinesin heavy and light chains are Kif5b and KLC2, respectively. Cargos have been known to bind to heavy chain either directly or indirectly via KLC[1,26]. By overexpression, we found that Dopey1 co-IPed KLC2 and co-expression of KLC2 was

required for Dopey1 to co-IP Kif5b (Supplementary Fig. 5a), therefore suggesting that Dopey1 probably interacts with Kif5b indirectly via KLC2.

Serial truncations of Dopey1 revealed that the N-terminal region consisting of ~150 residues, which was referred to as Dopey1 extreme N-terminal region (DEN), is necessary and sufficient to interact with co-expressed KLC2 (Fig. 5b and

**Fig. 3** Dopey1 specifically and directly interacts with PI4P. **a–f** The Golgi localization of Dopey1 and DEC, but not Mon2, is sensitive to PAO treatment. GFP-tdDEC transfected (**a, b**) or untransfected cells (**c–f**) were treated with 10 μM PAO followed by staining of indicated proteins. Dotted white lines demonstrate cell contours (**c, e**). In **b, d**, and **f**, the normalized Golgi intensity per cell was plotted against the time below the corresponding panel; error bar indicates mean ± s.e.m; n is the number of cells analyzed; results are representatives of three independent experiments. **g, h** The Golgi localization of Dopey1, but not Mon2, decreases upon the depletion of PI4KIIIβ. Indicated shRNAs were expressed using lentivirus and cells were stained for indicated proteins. The normalized Golgi intensity per cell is plotted at the right. Red dot, individual data point; error bar, mean ± s.d. from n = 3 independent experiments. **i–m** Dopey1 and DEC specifically interact with PI4P. Liposomes with 10% (m/m) indicated phosphoinositides were incubated with either the cell lysate expressing indicated fusion protein(s) (**i–l**) or purified recombinant protein (**m**). Proteins pulled down were analyzed by immunoblotting using indicated antibody. **n, o** Dimerized Dopey1 does not require Mon2 for its Golgi localization. Cells were subjected to control or Mon2 knockdown followed by transient expression of indicated construct. Endogenous Golgin97 (a Golgi marker) (**n**) or Golgin245 (**o**) was stained. Scale bar, 10 μm. Molecular weights (in kDa) are labeled in all gel blots. HeLa cells were used in (**a–h, n, o**), while HEK293T cells were used in (**i–l**). P values are from t test (unpaired and two-tailed). N.S. not significant, ***P ≤ 0.0005. Source data are provided as a Source Data file

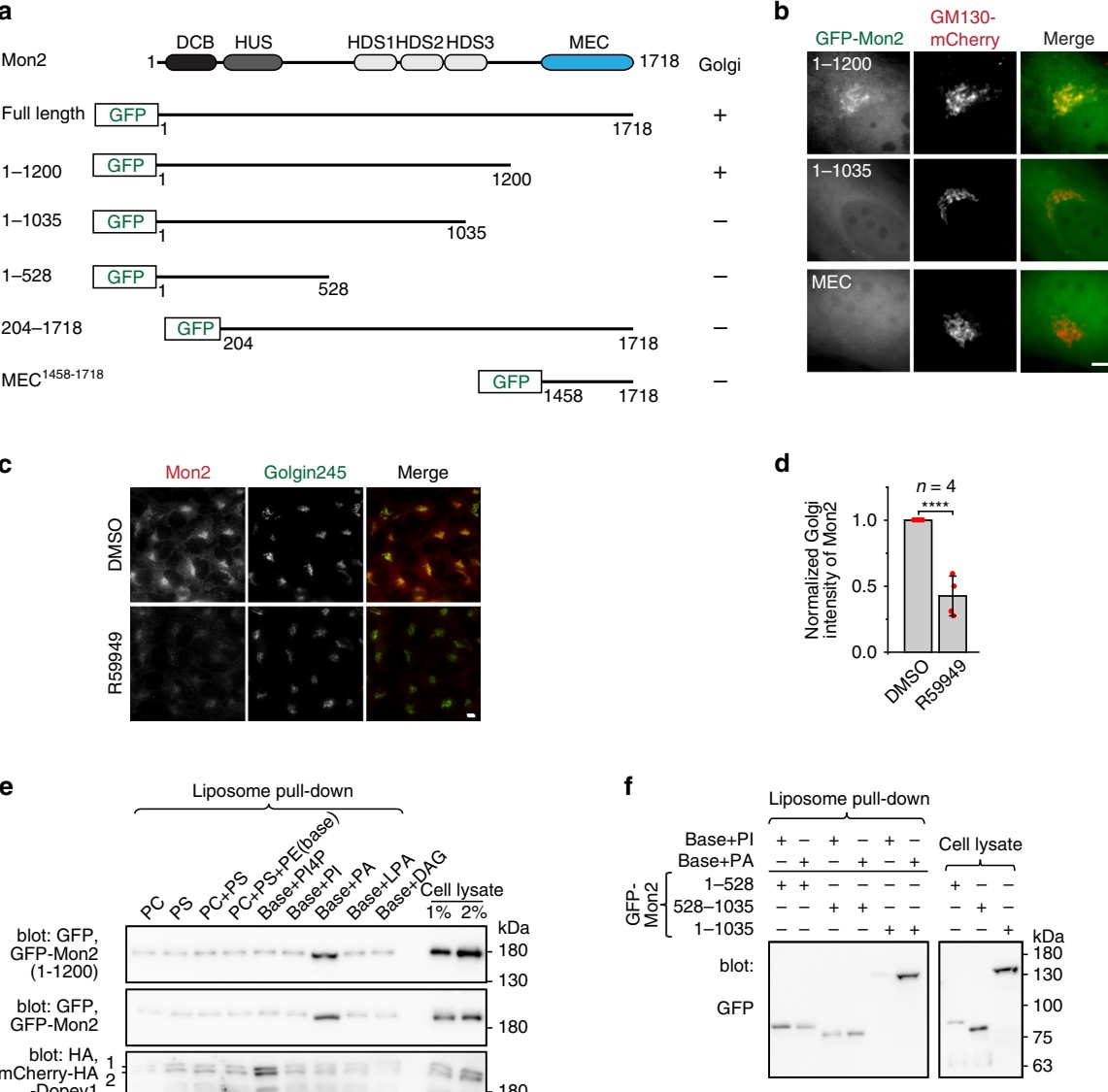

**Fig. 4** The Golgi localization of Mon2 requires its binding to PA. **a, b** The region comprising residues 1–1200 is necessary and sufficient for the Golgi localization of Mon2. Schematic diagrams of GFP-tagged Mon2 truncations are shown in (**a**). The Golgi localization is scored based on fluorescence staining experiment in (**b**), in which cells co-expressing selected Mon2 truncations and GM130-mCherry (a Golgi marker) were imaged live. **c** R59949 reduces the Golgi localization of Mon2. Cells were treated with DMSO (control) or 1 μM R59949 for 1 h followed by co-staining of endogenous Mon2 and Golgin245. Normalized Golgi intensity of Mon2 is plotted in (**d**). Red dot, individual data point; error bar, mean ± s.d. from four independent experiments. P values are from t test (unpaired and two-tailed); ****P ≤ 0.00005. **e, f** Mon2 specifically interacts with PA. The liposome pull-down experiments were similar to Fig. 3i–l except that liposomes with indicated phosphoglycerides were used. 1 and 2 indicate full length and a possibly degraded product of Dopey1 fusion protein. Molecular weights (in kDa) are labeled in gel blots. Scale bar, 10 μm. HeLa cells were used in (**b, c**), while HEK293T cell lysates were used in (**e, f**). Source data are provided as a Source Data file

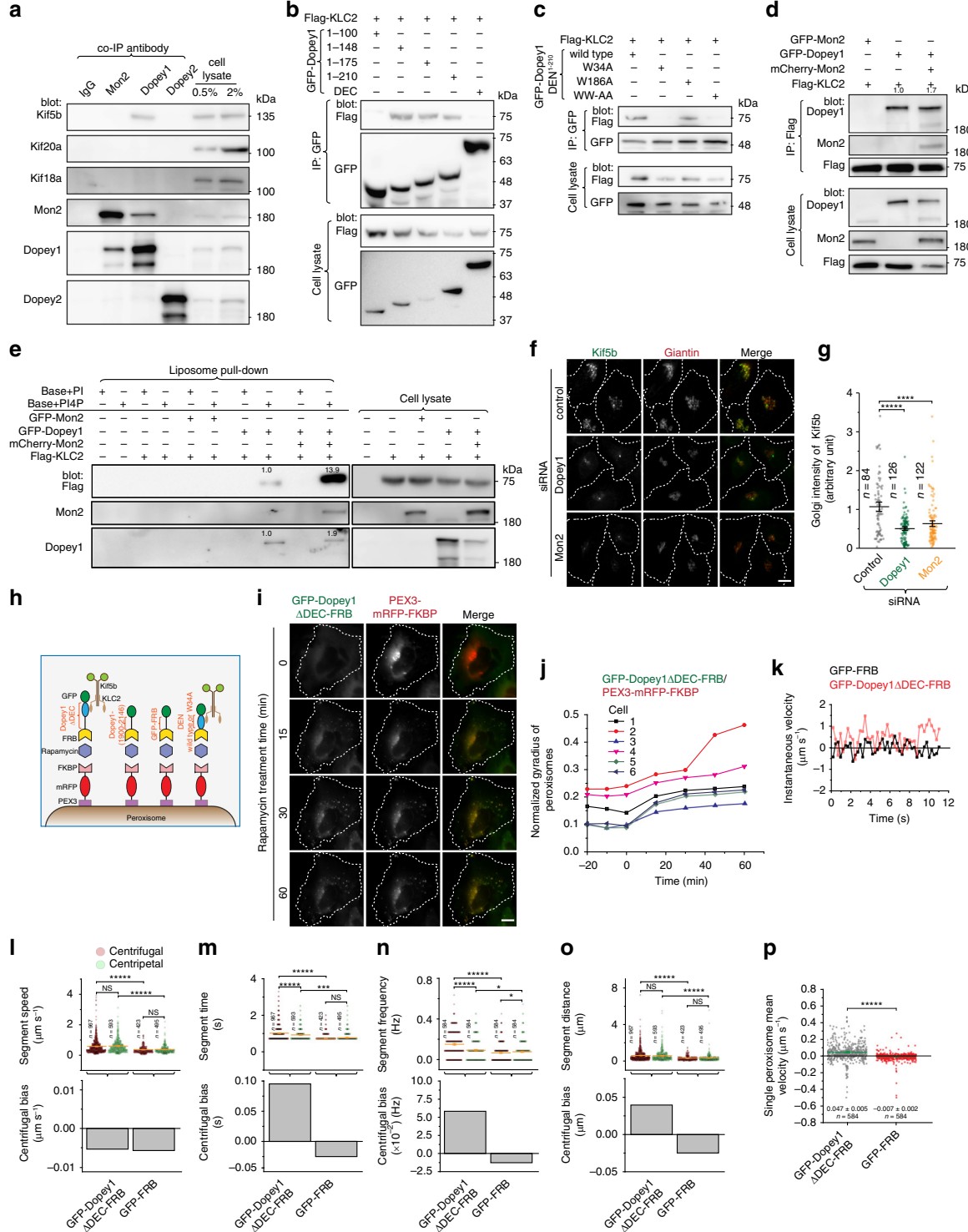

Supplementary Fig. 5a–c). The tetracopeptide repeat (TPR)
region of KLC2 was responsible for its direct binding to DEN,
as shown using in vitro translated and purified proteins
(Supplementary Fig. 5d), in agreement with what is known for
TPR as a cargo-binding domain of KLC[1]. Close examination of
the primary sequence of DEN revealed two tryptophan residues,
$W^{34}$ and $W^{186}$. $W^{34}$ is flanked by acidic residues ($EW^{34}AD$),
reminiscent of the previously reported KLC-binding tryptophan-
acidic (WD) motif[27]. The mutation of $W^{34}$, but not $W^{186}$, was
found to abolish the interaction between DEN and KLC2 (Fig. 5c;
Supplementary Fig. 5d), demonstrating that the single WD-motif

at $W^{34}$ is required for Dopey1 to recruit kinesin-1. Altogether, we
conclude that Dopey1 can interact with kinesin-1 via KLC2.

**Dopey1–Mon2 complex recruits kinesin-1 to the membrane.**
Since Dopey1 directly interacts with both Mon2 and KLC2, we
next asked if Dopey1, Mon2, and KLC2 can assemble into a
complex. Using cell lysates expressing transfected proteins, it was
observed that Flag-KLC2 co-IPed fluorescence protein (FP)-tag-
ged Mon2 only in the presence of GFP-Dopey1 (Fig. 5d),
implying that Dopey1 can link KLC2 and Mon2 to assemble into
a complex. Interestingly, we repeatedly observed that co-IPs

**Fig. 5** Dopey1 is sufficient to interact with and recruit kinesin-1 to the membrane. **a** Dopey1 specifically interacts with Kif5b. **b** Dopey1(1–148) is sufficient to interact with KLC2. **c** W$^{34}$ is essential for DEN's interaction with KLC2. **d** Dopey1–KLC2 interaction is enhanced by Mon2. Ratios of Dopey1 band intensity of the IP panel to that of the corresponding cell lysate panel were normalized and labeled in the top blot. IPs were similarly performed as in Fig. 1. **e** The membrane recruitment of KLC2 requires PI4P and Dopey1 and is enhanced by Mon2. Liposomes with PI or PI4P were incubated with cell lysates expressing indicated protein(s) and pull-downs were immunoblotted. Values are normalized ratios of band intensities in pull-down panels to corresponding ones in cell lysate panels. Molecular weights (in kDa) are labeled. **f, g** The Golgi association of Kif5b decreases upon the knockdown of Dopey1 or Mon2. The experiment and quantification are similar to Fig. 2d, e. **h–j** Artificially tethering Dopey1 is sufficient to move peroxisomes to the cell periphery. A schematic diagram of fusion proteins is shown in (**h**). **i** Cells co-expressing indicated proteins were treated with 50 nM rapamycin followed by live cell imaging. The normalized gyradius is quantified in (**j**). **k–p** Particle analysis of peroxisomes with artificially tethered Dopey1. **k** Instantaneous velocity of a typical peroxisome tethered with indicated fusion protein. Segment speed, time, frequency, distance, and their corresponding centrifugal biases are shown in (**l–o**). **p** Statistical analysis of single peroxisome mean velocity. In each case, tracks were identified from time lapses of seven cells in three experiments. *n* indicates the number of cells (**g**), segments (**l, m, o**), or tracks (**n, p**). Scale bar, 10 μm; error bar, mean ± s.e.m.; *P* values are from *t* test (unpaired and two-tailed); N.S. not significant, *$P \le 0.05$, ***$P \le 0.0005$, ****$P \le 0.00005$, *****$P \le 0.000005$. Dotted white lines indicate cell contours (**f, i**). HEK293T cells were used in **a-e** while HeLa cells were used in (**f, g, i–p**). Source data are provided as a Source Data file

between Dopey1 and KLC2 became slightly stronger in the presence of overexpressed Mon2 (Fig. 5d; Supplementary Fig. 5e), suggesting that Mon2 might enhance Dopey1–KLC2 interaction.

To investigate if Dopey1 can recruit kinesin-1 to the membrane, we employed the liposome pull-down experiment. Cell lysates containing various combinations of Flag-KLC2, GFP-Dopey1, and FP-Mon2 were incubated with base- or PI4P-liposome (Fig. 5e). It was found that Flag-KLC2 did not bind to PI4P-liposome unless GFP-Dopey1 was present. Though FP-Mon2 alone bound to neither base- nor PI4P-liposome, its presence substantially increased the recruitment of both Flag-KLC2 and GFP-Dopey1 to PI4P-liposome (Fig. 5e). From the disproportional increase in the recruited Flag-KCL2, Mon2-bound Dopey1 seemed to engage KLC2 more strongly, which was further substantiated in Supplementary Fig. 5e. Hence, our results suggest that Mon2 probably activates Dopey1 by dimerizing the latter to recruit kinesin-1 complex. In HeLa cells, endogenous Kif5b was mainly observed at the Golgi[28,29] (Fig. 5f) and its Golgi localization decreased greatly upon depletion of either Mon2 or Dopey1 (Fig. 5f, g), providing in vivo evidence that Dopey1 and Mon2 are essential for the Golgi recruitment of kinesin-1.

**Dopey1-tethering can centrifugally transport peroxisomes.** Kinesin-1 is a plus-end directed microtubule motor that can centrifugally transport cargos to the cell periphery[1,26]. Our discovery prompted us to examine if Dopey1 can function as a kinesin-1 adaptor. The hypothesis was tested by a rapamycin-induced peroxisome translocation assay[30]. Briefly, cells were co-transfected to express peroxisome-targeted PEX3-mRFP-FKBP together with FRB-fused Dopey1 fragment (Fig. 5h). Upon rapamycin-induced heterodimerization between FKBP and FRB, FRB-chimeras were forced to associate with the peroxisomal membrane and the ensuing dynamics of peroxisomes was imaged live. To quantitatively describe the ensemble movement of peroxisomes, we adopted the concept of gyradius from mechanics. The gyradius of the organelle in a cell is calculated as the intensity-weighted root mean square of pixel-distances to the center of fluorescence mass (see Methods). The differences in cell sizes and shapes are normalized by the gyradius of the cell edge. The normalized gyradius quantitatively depicts the overall distribution of the organelle within a cell.

In HeLa cells, peroxisomes are mostly centrally positioned (0 min of Fig. 5i, Supplementary Fig. 5f, g and Supplementary Movie 1) and the normalized gyradius of peroxisomes in individual cells remained constant over time, as expected (−20 to 0 min in Fig. 5j). Both Dopey1ΔDEC (Fig. 5i; Supplementary Movie 1) and Dopey1(1900–2146) (Supplementary Fig. 5 h) were cytosolic due to their lack of DEC. Upon the administration of rapamycin, both were rapidly recruited to peroxisomes; however,

DEN-containing Dopey1ΔDEC (Fig. 5i; Supplementary Fig. 5f, g; Supplementary Movie 1), but not DEN-missing Dopey1 (1900–2146) (Supplementary Fig. 5h, i), dispersed peroxisomes, as quantitatively reflected by their normalized gyradii. Further observation that wild type DEN, but not W34A, alone was able to disperse clustered peroxisomes (Supplementary Fig. 5j–m; Supplementary Movie 2) led us to conclude that DEN is necessary and sufficient for Dopey1-mediated centrifugal movement upon artificially tethering to peroxisomes.

To understand the movement of individual peroxisomes, we employed high spatial and temporal resolution imaging coupled with particle tracking to isolate trajectories of single peroxisomes (see Methods). We used the polar coordinate system with its origin at the cell's center of the fluorescence mass and defined centrifugal direction as positive. Dopey1ΔDEC-tethered peroxisomes were found to display more dynamic bidirectional movement than GFP-FRB-tethered ones (Fig. 5k), with an increase in not only the centrifugal segment parameters but also centripetal ones (Fig. 5l–o). Since the relative differences between the two opposite direction parameters rather than their absolute values determine the net transport, we defined a quantity, the centrifugal bias of a segment parameter, as the centrifugal segment parameter minus the corresponding centripetal one. It becomes obvious that tethering Dopey1 increases the centrifugal bias of all segment parameters (Fig. 5l–o). Indeed, statistics revealed that single peroxisome mean velocity within the time span of our imaging of Dopey1ΔDEC-tethered peroxisomes is 0.047 μm s$^{-1}$, while that of GFP-FRB-tethered peroxisomes is −0.007 μm s$^{-1}$ (Fig. 5p). The significant increase in the centrifugal speed is consistent with the Dopey1-mediated centrifugal movement of peroxisomes. Taken together, our results showed that Dopey1 can function as a kinesin-1 adaptor.

**Periphery positioning of organelles requires Dopey1–Mon2.** Under live cell imaging, FP-tagged Dopey1 (Supplementary Fig. 6a) and Mon2 (Fig. 6a) also displayed many puncta in addition to their Golgi localization. Dopey1 and Mon2 puncta were observed to colocalize with each other and organelles in secretory and endocytic pathways, such as the EE (mCherry-Rab5), EE/RE (internalized transferrin- or Tf-594), LE (mCherry-Rab7), lysosome (Lamp1-mCherry), and ERES/ERGIC (Sec31a-mCherry). The localization to the RE and lysosome can also be noted for endogenous Mon2 despite the high background noise (Fig. 6b). However, Dopey1 and Mon2 were not found at the peroxisome and mitochondrion (Supplementary Fig. 6b). The ERES/ERGIC and endolysosomal localization of Dopey1 and Mon2 was sensitive to brefeldin A, while only that of Dopey1 was sensitive to PAO (Supplementary Fig. 6c, d), consistent with our

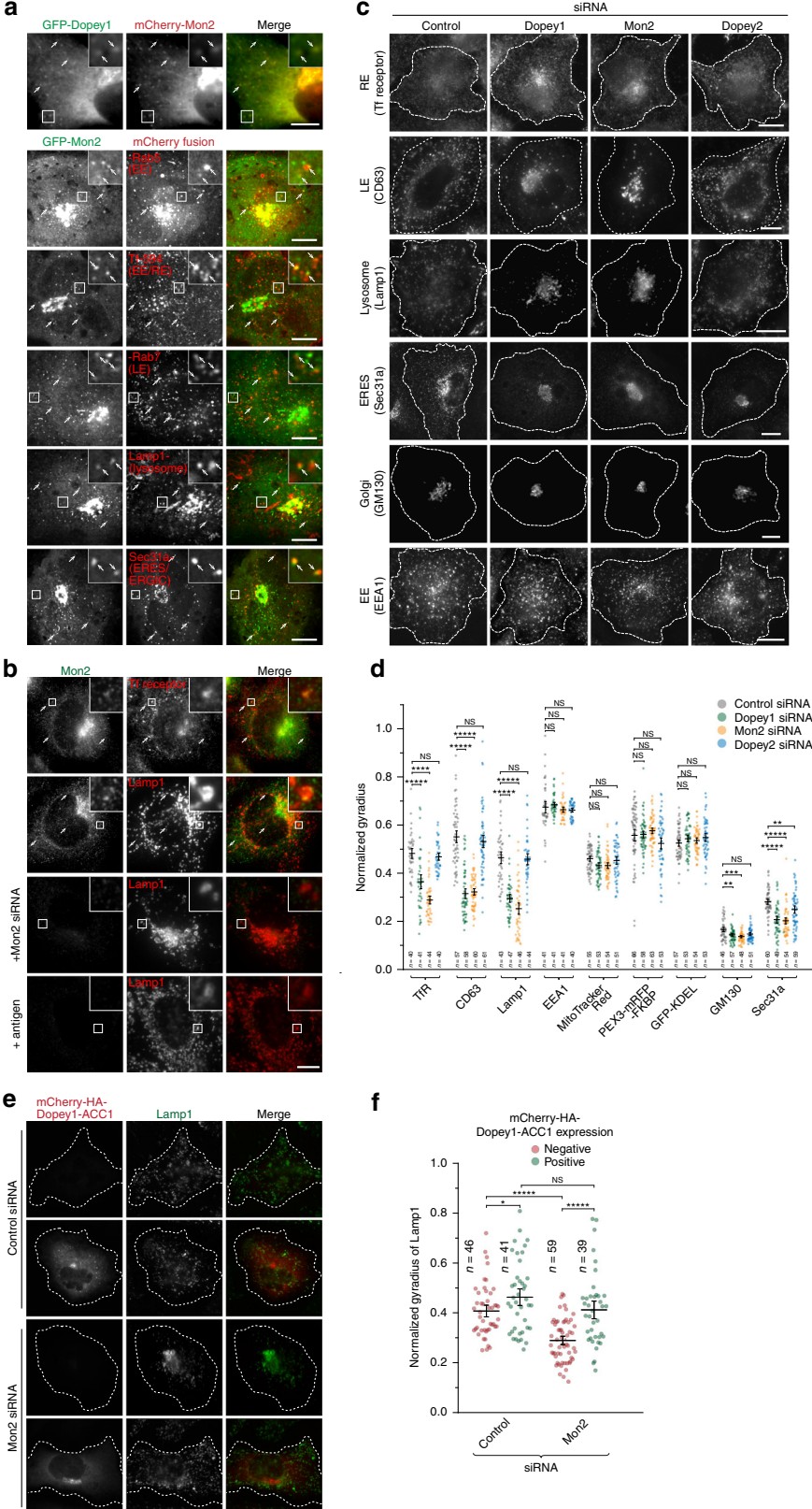

model that the membrane association of Mon2 is dependent on Arf1 and that of Dopey1 requires both Mon2 and PI4P.

To test if Dopey1–Mon2 complex functions in organelle positioning, we investigated the subcellular distribution of various organelles when endogenous Mon2, Dopey1, or Dopey2 was depleted (Fig. 6c; Supplementary Fig. 6e). In control and Dopey2-depleted cells, the RE, LE, lysosome, and ERES/ERGIC (except Dopey2-depletion here) were observed to spread from the cell center to periphery. In contrast, when either Mon2 or Dopey1 was depleted, they became clustered around the cell center. The aggregation of these organelles, such as the RE and lysosome, required the microtubule network (Supplementary Fig. 6f, g).

**Fig. 6** Dopey1–Mon2 complex is essential for the peripheral positioning of organelles. Hela cells were used. **a** FP-Mon2 colocalizes with FP-Dopey1 at puncta and localizes to the EE/RE, LE, lysosome, and ERES/ERGIC in live cell imaging. Cells transiently expressing indicated FP-tagged proteins were imaged live under a wide-field (the first row) or spinning disk confocal microscope (the rest rows). The EE/RE was labeled by pulse-labeling cells with 30 μg ml$^{-1}$ Tf-594 for 5 min followed by 20 min chase. **b** Endogenous Mon2 displays punctate staining pattern and partially colocalizes with the EE/RE (Tf receptor) and lysosome. Cells with or without Mon2 siRNA treatment were stained for indicated endolysosomal markers and Mon2. In the last row, cells were stained by antigen-neutralized Mon2 antibody. The region of interest is enlarged at the upper-right corner. Arrows indicate colocalization. **c**, **d** Depletion of Dopey1 or Mon2, but not Dopey2, causes the aggregation of various secretory and endocytic organelles at the cell center. After treatment with indicated siRNAs, cells were processed for immunofluorescence labeling of indicated organelle markers. Alternatively, cells were transfected to express PEX3-mRFP-FKBP or GFP-KDEL to label the peroxisome or ER, respectively. Mitochondria were labeled by MitoTracker Red. Typical images are shown in (**c**), while normalized gyradii are plotted in (**d**). **e**, **f** Homodimerized Dopey1 can rescue Mon2-depletion-induced lysosomal aggregation. siRNA-treated cells were transfected to express Dopey1-ACC1 followed by immunostaining of Lamp1. Typical images and normalized gyradii are shown in (**e**, **f**), respectively. **d**, **f** Representative results of three independent experiments. Dotted white lines indicate cell contours (**c**, **e**). Scale bar, 10 μm; error bar, mean ± s.e.m.; $n$ the number of cells analyzed, $P$ values are from $t$ test (unpaired and two-tailed), N.S. not significant, *$P \le 0.05$, **$P \le 0.005$, ***$P \le 0.0005$, ****$P \le 0.00005$, *****$P \le 0.000005$. Source data are provided as a Source Data file

Low-cytoplasmic pH is known to cause the kinesin-1-dependent peripheral distribution of lysosomes[31], of which the molecular mechanism is still unclear. Using Ringer's acetate solution (pH6.5), we demonstrated that Dopey1 and Mon2 are required for acidification-induced centrifugal positioning of lysosomes (Supplementary Fig. 6h).

Although Dopey1 and Mon2 can be detected on the EE (Fig. 6a and Supplementary Fig. 6a), the peripheral distribution of the EE seemed unaltered upon their knockdown (Fig. 6c), probably due to the dominant action by motors such as Kif16b[32]. Their depletion did not change the positioning or the morphology of the mitochondrion, peroxisome, and ER either (Supplementary Fig. 6e, i). These changes upon knockdowns were also quantitatively reflected by normalized gyradii of organelle markers (Fig. 6d). Interestingly, although the Golgi always resides at the cell center, it appeared to shrink and spread less upon the depletion of either Dopey1 or Mon2 (Fig. 6d; Supplementary Fig. 6e). The effect on organelle positioning was rescued by exogenously expressing RNAi-resistant Dopey1 or Mon2 (Supplementary Fig. 6j–m). We found that Dopey1-ACC1, which is not targeted by Dopey1 siRNA, rescued the lysosome positioning; in contrast, Dopey1-ACC1 with W34A or LKRL-4A mutation failed (Supplementary Fig. 6n, o). Dopey1-ACC1 was also able to rescue the lysosome positioning phenotype caused by Mon2 depletion (Fig. 6e, f). In summary, our data demonstrated that Dopey1–Mon2 complex is essential for the peripheral positioning of the RE, LE, Lysosome, ERES/ERGIC and the expansion of the Golgi, consistent with its function as a kinesin-1 adaptor.

**Dopey1 and Mon2 promote centrifugal membrane trafficking.** Besides organelle positioning, kinesin-1 also functions in the bidirectional transport of membrane carriers in secretory and endocytic pathways[1]. We systematically investigated the effect of depleting Dopey1 or Mon2 on the membrane trafficking between the centrally localized Golgi and organelles at the periphery, including the PM and ER.

To study the Golgi-to-PM or post-Golgi secretory trafficking, we employed 20 °C block[33] to synchronize the ER-released RUSH (retention using selective hooks) reporters[34], including E-cadherin and TNFα (Supplementary Fig. 7a), at the Golgi. We found that the depletion of Dopey1 or Mon2 significantly reduced the Golgi-to-PM trafficking (Fig. 7a; Supplementary Fig. 7b, c). Interestingly, while the Golgi-pool of reporters quickly emptied in control cells, it persisted much longer in Dopey1 or Mon2 knockdown cells (Fig. 7a–c; Supplementary Fig. 7d), demonstrating that Dopey1 and Mon2 are essential for the Golgi exit of cargos, probably by facilitating the biogenesis of membrane carriers. In live cell imaging, we observed that FP-tagged Dopey1, Mon2, Kif5b, P4M, and Rab6 were present in Golgi-derived and reporter-containing carriers (Fig. 7d; Supplementary Fig. 7e), suggesting that Dopey1–Mon2 complex probably can assemble on the PI4P-containing and Rab6-positive post-Golgi carriers to recruit kinesin-1 motor for centrifugal trafficking.

Similar to our study on peroxisomes, individual carriers released form the Golgi were subjected to single-particle analysis (Fig. 7e–j). Single carriers displayed bidirectional movement in both control and Dopey1 or Mon2 depleted cells. Compared to the control, depletion of Dopey1 or Mon2 decreased the centrifugal bias of segment parameters (Fig. 7f–i). As expected, the single carrier mean velocity decreased from 0.133 μm s$^{-1}$ in control cells to around 0.03 μm s$^{-1}$ in Dopey1 and Mon2 depleted cells (Fig. 7j). In summary, our data support a model according to which dual lipids, PI4P and PA, might be responsible for the membrane association of Dopey1–Mon2 complex, which in turn recruits kinesin-1 to promote the biogenesis and periphery trafficking of carriers.

We previously reported that the depletion of Mon2 accelerates the endocytic trafficking of CD8a-furin and -CI-M6PR to the Golgi[5]. Our further observation revealed that the depletion of Dopey1, but not Dopey2, produced similar acceleration effect on CD8a-furin, -sortilin and -CI-M6PR (Fig. 7k, l; Supplementary Fig. 7f–h). Different from the Golgi-to-PM trafficking[35], the PM-to-Golgi trafficking is known to involve various types of endosomes as intermediates, such as the EE, RE, and LE[36], which are centripetally positioned around the Golgi in Dopey1 or Mon2 depletion cells. Hence, our observation might be accounted for by the accelerated centripetal transport of carriers in the PM-to-endosome and/or endosome-to-Golgi pathway. Though we are unable to resolve these mechanisms, our data demonstrated that Dopey1–Mon2 complex probably impedes the PM-to-Golgi trafficking.

When ManII, a RUSH reporter and Golgi resident membrane protein, and VSVGtso45, a temperature sensitive exogenous membrane protein that targets to the PM, were synchronously released from the ER, the two reporters were rapidly transported to the Golgi. Two types of ManII puncta were observed (Supplementary Fig. 8a). Those colocalizing with Sec31a were probably transiting the ERES/ERGIC and were largely immobile; those not colocalizing were mobile and likely transport carriers under bidirectional movement toward the Golgi (Supplementary Fig. 8a). We observed that different cargos, such as ManII and mCherry-GPI, were transported in the same mobile puncta (Supplementary Fig. 8b). Whether these puncta are ERGICs or ERGIC-derived anterograde transport carriers is still in debate[37]. Regardless of the cellular identity of these puncta, we observed that depletion of Dopey1 or Mon2 accelerated the transport of ManII and VSVGtso45 to the Golgi (Fig. 8a–c; Supplementary Fig. 8c). In live cell imaging, FP-tagged Dopey1 and Mon2 were

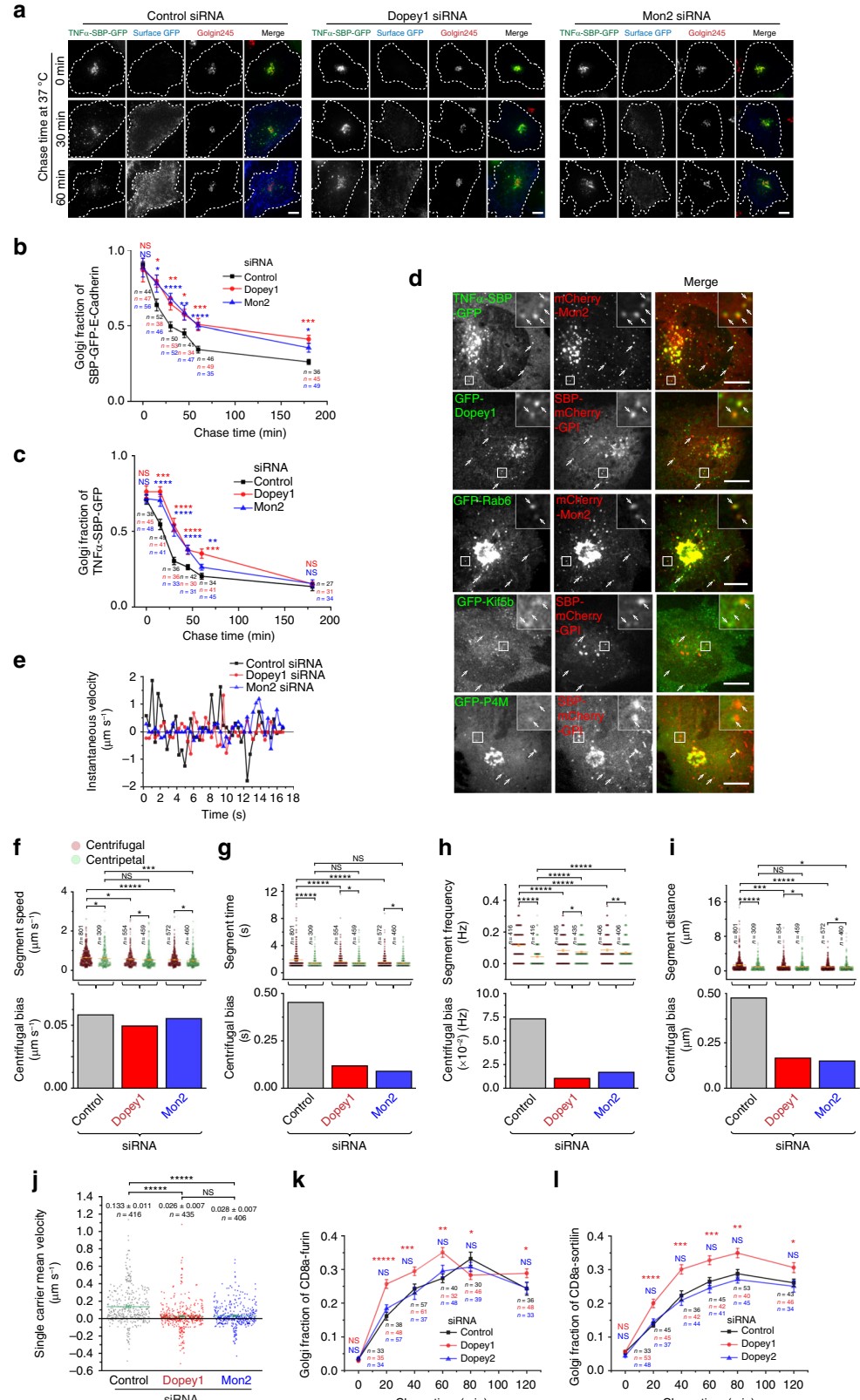

observed in mobile and cargo-positive transport carriers in additional to those immobile ones (Fig. 8d, e), which are likely ERESs/ERGICs.

Our particle analysis showed that cargo-positive carriers, which predominantly move centripetally toward the Golgi, also appeared bidirectional movement in both control and

knockdown cells (Fig. 8f–j; Supplementary Fig. 8d). Interestingly, the depletion of Dopey1 or Mon2 resulted in more dynamic bidirectional movement by increasing all three segment parameters in not only centripetal but also centrifugal direction (Fig. 8f–i). The single carrier mean velocity changed from −0.055 µm s⁻¹ in control to −0.126 and −0.147 µm s⁻¹ in Dopey1 and

**Fig. 7** Dopey1–Mon2 complex promotes the Golgi-to-PM but inhibits the reverse trafficking. **a–c** The Golgi exit of cargos and Golgi-to-PM trafficking require Dopey1 and Mon2. siRNA-treated cells expressing indicated RUSH reporter were subjected to the Golgi-to-PM trafficking assay (see Methods). Reporters arrived at the PM were surface-labeled by anti-GFP (**a**). Dotted white lines indicate cell contours. Golgi fractions were plotted in (**b**, **c**). **d** Mon2, Dopey1, Rab6, Kif5b, and PI4P can be found on membrane carriers involved in the Golgi-to-PM trafficking. Cells co-expressing indicated proteins were subjected to biotin and 20 °C treatment similar to **a**, and subsequently imaged live during 37 °C chase under a spinning disk confocal microscope. The region of interest is enlarged at the upper right. Arrows indicate colocalization. **e–j** Particle analysis of single carriers in the Golgi-to-PM trafficking. siRNA-treated cells expressing SBP-GFP-E-cadherin were subjected to treatment described in (**d**) except that imaging was conducted under the wide-field microscopy. See Supplementary Movie 3 for the control. **e** Instantaneous velocity of a typical carrier. Segment speed, time, frequency, distance, and their corresponding centrifugal biases are displayed in (**f–i**). **j** The statistical analysis of single carrier mean velocity. n, the number of segments in (**f**, **g**, **i** or tracks in **h**, **j**). In each case, tracks were identified from time lapses of eight cells in three experiments. **k**, **l** Depletion of Dopey1 or Mon2 accelerates the PM-to-Golgi trafficking. siRNA-treated cells expressing indicated CD8a-TGN-tails were subjected to the PM-to-Golgi trafficking assay (see Methods) and Golgi fractions were plotted against the chase time. Scale bar, 10 μm; error bar, mean ± s.e.m. In **b**, **c**, **k** and **l**, plots are representative results of three independent experiments; n indicates the number of cells analyzed. P values are from the t test (unpaired and two-tailed) comparison between indicated pairs (**f–j**) or between the knockdown and corresponding control (**b**, **c**, **k**, **l**). N.S. not significant, $*P \leq 0.05$, $**P \leq 0.005$, $***P \leq 0.0005$, $****P \leq 0.00005$, $*****P \leq 0.000005$. Source data are provided as a Source Data file

Mon2 knockdown cells, respectively (Fig. 8j), demonstrating significant increases in the net transport speed in centripetal direction or toward the Golgi. Therefore, our data demonstrated that Dopey1–Mon2 complex probably impedes the ER-to-Golgi trafficking.

GFP-ERGIC53, a membrane protein that continuously cycles between the ER and Golgi, was employed to investigate the Golgi-to-ER or retrograde trafficking. We took advantage of the fluorescence recovery after photo-bleaching (FRAP) technique to selectively photo-bleach the ER-pool of GFP-ERGIC53. The fluorescence recovery of the ER-pool therefore indicated the steady-state Golgi-to-ER trafficking of GFP-ERGIC53. The recovery was observed to be much slower in Dopey1 and Mon2 knockdown than control cells (Fig. 8k, l), therefore suggesting that Dopey1–Mon2 complex probably promotes the Golgi-to-ER trafficking.

Since the Golgi are centrally localized while the PM and ER are peripherally localized, when all our results are taken altogether, we conclude that, as summarized in Fig. 9a, Dopey1–Mon2 complex promotes centrifugal and inhibits centripetal transport of membrane compartments or carriers.

## Discussion

Both Dopey1 and Mon2 are poorly studied previously. Here, we gave a detailed molecular and cellular characterization of both proteins. Based on our data, we propose a model in which Dopey1 and Mon2 assemble into a complex that coincidentally detects dual-lipids, PI4P and PA (directly or indirectly via other factor), and recruits kinesin-1 motor via KLC2 (Fig. 9b). Note that in addition to PA, the membrane association of Mon2 seems to require active Arf1[5]. We found that Dopey1–Mon2 complex is essential for the centrifugal positioning or expansion of organelles; it promotes centrifugal but inhibits centripetal trafficking pathways (Fig. 9a). This represents a kinesin-1 adaptor for generic membrane trafficking and organelle positioning. Alternative kinesin motor systems probably co-exist on the same membrane compartment for the bidirectional transport. This has been especially well illustrated for the lysosome. For instances, multiple kinesin-1 adaptors, such as Ragulator-BORC1-Arl8-SKIP[38–40] and Rab7-FYCO1[41], have been found on the lysosome. In addition to kinesin-1, a kinesin-13 family motor, KIF2β, was also discovered there[42]. It is currently unclear about the molecular and cellular significance of multiple kinesins on the same lysosome. Perhaps they localize to different subpopulations of lysosomes or provide redundancy to ensure the robustness of lysosomal transport system.

Both PI4P and PA are important structural and signaling components of the membrane and their local concentrations are dynamically regulated by opposing enzymes[9,10,18,19,25]. Though predominantly enriched in the Golgi, PI4P has also been detected in multiple organelles or domains of organelles, such as the PM[11], endolysosome[14], and ERES[43,44]. The presence of PA at these localizations was also documented[45–49]. Hence, it might be reasonable to assume that transport carriers derived from these compartments also contain PI4P and PA. Consistent with this notion, our imaging data verified the localization of Dopey1 and Mon2 at the Golgi, endolysosome, ERES/ERGIC and membrane carriers derived from the Golgi and ERES/ERGIC (Figs. 2a, 6a, 7d and 8d, e; Supplementary Fig. 6a and 7e). Dopey1–Mon2-kinesin-1 motor system likely provides a driving force for the centrifugal movement of these organelles, therefore accounting for its essential role in the centrifugal trafficking or positioning.

PA has been proposed to generate membrane curvature[50] and is known to facilitate the biogenesis of COPI[51] and COPII-coated vesicles[45]. On the other hand, PI4P participates in cargo sorting and clathrin coat assembly[52] as well as the ER export and post-Golgi constitutive trafficking[43,53–55]. Our data further uncovered that PI4P and PA might serve as a dual-lipid platform to recruit kinesin-1 motor, which provides mechanical force for the proper morphology of the Golgi, the biogenesis of membrane carriers and their subsequent transport along the microtubule tracks. Combining with findings from other labs, it is tempting to propose that, together with Rab6[56], PI4P and PA probably define the recently discovered post-Golgi carrier fission hot spot by recruiting kinesin and myosin motors.

It has been established that the intracellular cargo movement is bidirectional and the overall net directionality is achieved through rounds of forward and backward movement[57–59]. Our single particle analysis revealed the role of Dopey1–Mon2 complex in promoting the centrifugally biased bidirectional transport. An interesting observation also emerged from our particle analysis of the dispersion of tethered-peroxisomes (Fig. 5l–o) and ER-to-Golgi trafficking (Fig. 8f–i). We noted that increasing a segment parameter at the ensemble net transport direction, which is larger than that of the opposite direction, appears to be counteracted by the corresponding increase in that of the opposite direction, resulting in a blunted change of the centrifugal bias. It has been previously reported in multiple systems that the inhibition of a motor decreases the opposite motor's activity[57], therefore resulting in a similarly reduced change of the centrifugal bias. Such effect probably makes the trafficking more stable as the ensemble net transport can withstand cellular perturbations. Hence, our observations support the previously proposed communication and cooperation between opposite motors[57–59].

Dopey1 has been proposed to be the causative gene behind the myelin pathology of vacuole formation (VF) rats, which possess a

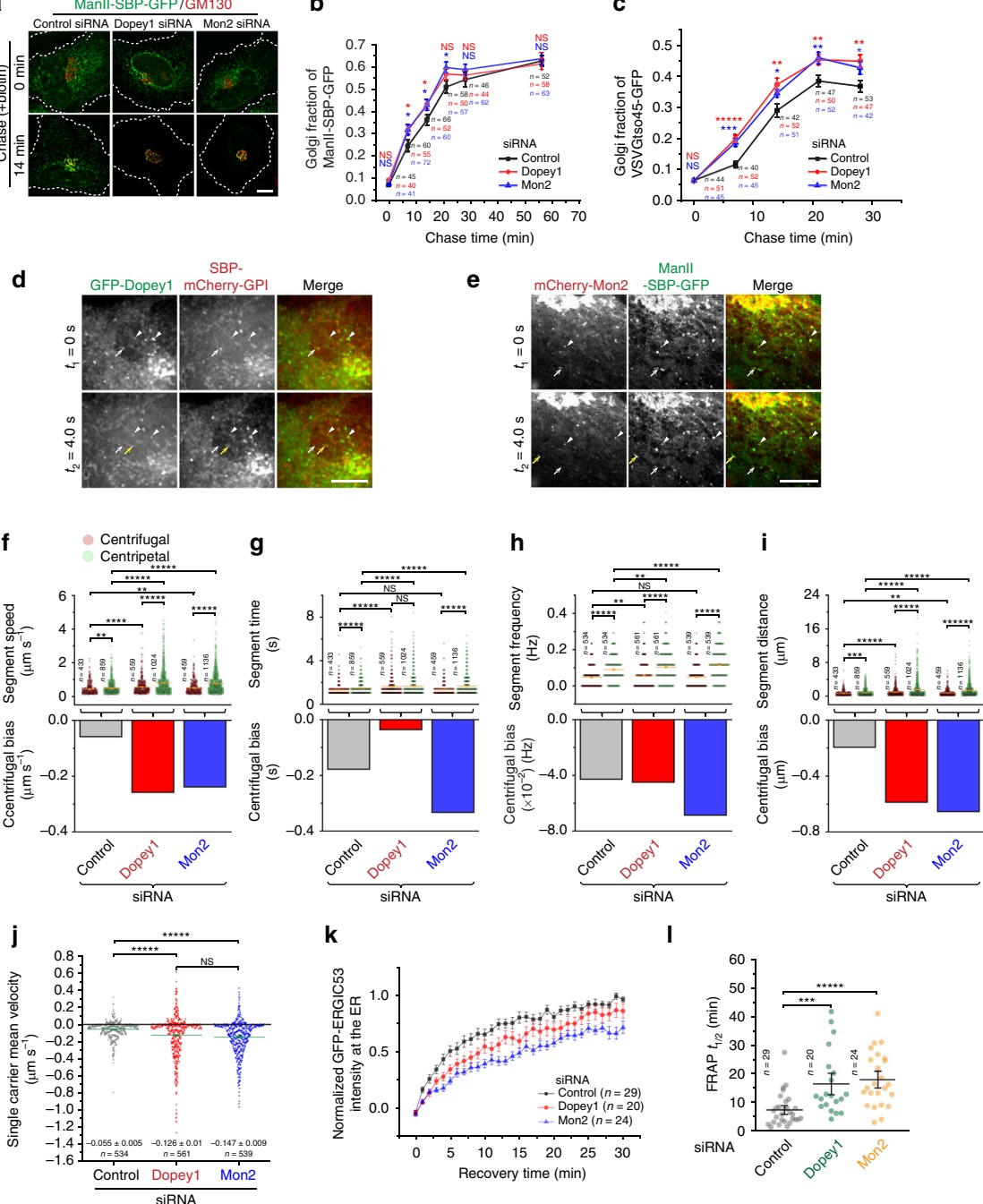

**Fig. 8** Dopey1–Mon2 complex inhibits the ER-to-Golgi but promotes the reverse trafficking. **a–c** The ER-to-Golgi trafficking requires Dopey1 and Mon2. siRNA-treated cells expressing ManII-SBP-GFP or VSVGtso45-GFP were chased in biotin or by temperature shift followed by staining of GM130. Deconvolved images are shown in (**a**). Dotted white lines indicate cell contours. Golgi fractions are quantified in (**b**, **c**). Plots are representatives of three experiments; $n$ indicates the number of cells analyzed. **d**, **e** Dopey1 and Mon2 can be found on mCherry-GPI- and ManII-positive carriers moving from the ER to the Golgi. Cells co-expressing indicated reporters were treated with biotin and the time lapse was acquired under a spinning disk confocal microscope. White and yellow arrows indicate positions of mobile carriers at time $t_1$ and $t_2$, respectively. Arrow heads show colocalization at immobile puncta. **f–j** Particle analysis of single carriers involved in the ER-to-Golgi trafficking. siRNA-treated cells transiently expressing ManII-SBP-GFP were chased with biotin during the time lapse imaging. See Supplementary Movie 4, for an example of the control. Segment speed, time, frequency, distance and their corresponding centrifugal biases are shown in (**f–i**). **j** The statistical analysis of single carrier mean velocity. $n$ the number of segments (**f**, **g**, **i**) or tracks (**h**, **j**). Tracks were identified from time lapses of 9, 7, and 9 cells in 3 experiments, respectively. **k**, **l** The Golgi-to-ER trafficking requires Dopey1 and Mon2. siRNA-treated cells expressing GFP-ERGIC53 were subjected to FRAP (see Methods). See Supplementary Movie 5, for an example of the control. Normalized ER intensity traces are in (**k**) and the FRAP $t_{1/2}$ of each experiment is shown in (**l**). Scale bar, 10 μm; error bar, mean ± s.e.m. $P$ values are from the $t$ test (unpaired and two-tailed) comparison of indicated pairs (**f–j**, **l**) or between the knockdown and corresponding control (**b**, **c**). N.S. not significant, *$P ≤ 0.05$, **$P ≤ 0.005$, ***$P ≤ 0.0005$, ****$P ≤ 0.00005$, *****$P ≤ 0.000005$. Source data are provided as a Source Data file

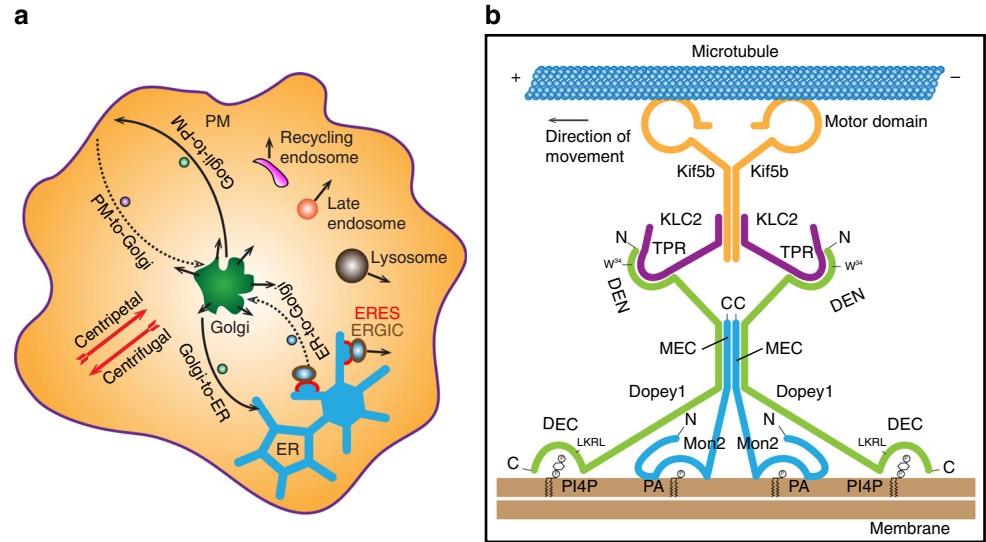

**Fig. 9** Schematic summary and model. **a** The diagram schematically summarizing the cellular function of Dopey1–Mon2 complex: it promotes the centrifugal (solid line with arrow) and inhibits the centripetal (dotted line with arrow) membrane trafficking and organelle positioning. **b** A schematic model explaining how Dopey1–Mon2 complex engages the membrane and kinesin-1. Mon2 is a constitutive homodimer and its dimerization is mediated by MEC at its C-terminus. Its N-terminal region interacts with PA (directly or indirectly). In contrast, Dopey1 is a monomer. However, two copies of Dopey1 can simultaneously bind to a Mon2 homodimer, resulting in the formation of a heterotetrameric complex. The interaction between Dopey1 and Mon2 is through Mon2's MEC and Dopey1's central region. DEN at the N-terminus of Dopey1 interacts with KLC2's TPR domain via the WD-motif, while DEC at the C-terminus functions as a PI4P-binding module. In Dopey1–Mon2 complex, Mon2 seems to dimerize Dopey1 for its high affinity binding to membrane-PI4P and KLC2 homodimer

nonsense mutation in Dopey1 and display hypomyelination and vacuolation in the central nervous system[60]. In the central nervous system, the myelin sheath of an axon is assembled from the peripheral cell extension of the oligodendrocyte. In oligodendrocytes of VF rats, proteolipid protein and myelin-associated glycoprotein abnormally accumulated in the Golgi within the cell body, instead of targeting to peripheral myelin sheath. In light of our discovery, we propose that such accumulation might be primarily due to the disruption of Dopey1–Mon2–kinesin-1-mediated post-Golgi carrier biogenesis and the subsequent centrifugal trafficking to the periphery myelin sheath. Therefore, the observation from VF rats demonstrates that Dopey1–Mon2 complex has potential physiological and pathological roles in the nervous system.

## Methods
**DNA plasmids**. Please see Supplementary Table 1.

**siRNAs**. GL2 control siRNA (CGUACGCGGAAUACUUCGA), Mon2 siRNA #1 (GGCAGUGGGUCAACCUUUA) and #2 (AAAUAUUGAUGUCGAGGUA), Dopey1 siRNA #1 (GCUAGGACCUGUAAUUCUA) and #2 (GCAGUGAUAUUGAGCUAAU), and Dopey2 siRNA #1 (GAACAAGGCUGGAAGCUCU) and #2 (GAACUCGGCCGAUGACUUG) were purchased from Dharmacon. Both #1 and #2 siRNAs targeting Dopey1 or Mon2 worked equally well with regards to phenotypes reported in this study. #1 siRNAs of Mon2, Dopey1 and Dopey2 were used for figure data except Supplementary Fig. 7f–h.

**Antibodies**. Mon2 rabbit polyclonal antibody (pAb) was raised and purified using a recombinant C-terminal fragment of Mon2[5]. Mouse monoclonal antibodies (mAbs) against Myc (#sc-40)(1:1000 for Western blot or WB), HA (#sc-7392) (1:1000 for WB), GFP (#sc-9996)(1:1000 for WB; 1:500 for immunofluorescence or IF), GAPDH (#sc-25778)(1:1000 for WB), GST (#sc-1382) (1:1000 for WB), Kif18a (#390600) (1:1000 for WB) and Kif20a (#sc-374508)(1:1000 for WB) were from Santa Cruz; mouse mAbs against GM130 (#610823)(1:100 for IF), Golgin245 (#611280)(1:100 for IF), GS28 (#611184)(1:250 for IF) and EEA1 (#610456)(1:500 for IF) were from BD Bioscience; rabbit pAb against giantin (#ab24586)(1:1000 for IF), mouse mAb against Kif5b (SUK4; #ab28060)(1:1000 for WB; 1:500 for IF) and horseradish peroxidase-conjugated Protein A (#ab7456)(1:3000 for WB) were from Abcam. Mouse mAb against Golgin97 (CDF4; #A-21270)(1:250 for IF) and Alexa Fluor-conjugated goat antibodies against mouse(1:500 for IF) IgG and rabbit IgG

(1:500 for IF) and IgM(1:500 for IF) were from Thermo Fisher Scientific. Mouse mAb against PI4P (#Z-P004)(1:100 for IF) was from Echelon Biosciences. Mouse mAb against Flag (#F-1804)(1:1000 for WB) was from Sigma-Aldrich. Mouse mAbs against transferrin receptor (OKT9)(1:250 for IF), CD63 (H5C6)(1:20 for IF), Lamp1 (H4A3)(1:500 for IF) and CD8a (OKT8)(1:250 for IF) were used in the form of hydridoma culture media, which were purchased from Developmental Studies Hybridoma Bank. Mouse anti-Sec31a(1:100 for IF) antibody was a gift from W. Hong. Horseradish peroxidase-conjugated mouse mAb against Myc (#A-00863)(1:3000 for WB) was from Genscript.

**Chemicals**. MitoTracker Red CMXRos and Tf-Alexa Fluor 594 was from Thermo Fisher Scientific. PAO and nocodazole were from Merck. Rapamycin, cycloheximide, and R59949 were from Sigma-Aldrich. Brefeldin A was from Epicenter Technologies.

**Cell culture and transfection**. HeLa and HEK293T cells were from American Type Culture Collection. 293FT cells were from Thermo Fisher Scientific. Cells were cultured in Dulbecco's Modified Eagle's Medium supplemented with 10% fetal bovine serum. Transfection of plasmid DNA and siRNA was conducted using polyethylenimine (Polysciences) or Lipofectamine2000 (Thermo Fisher Scientific), according to manufacturer's protocol. All cells used in imaging are HeLa cells. All cells used in IPs, pull-downs, and immunoblots are HEK293T cells unless specified otherwise.

**Lentivirus-mediated knockdown of PI4KIIIβ**. GL2 control shRNA, PI4KIIIβ shRNA#1 or #2 in pLKO.1 vector was transiently transfected to 293FT cells together with pLP1, pLP2, pLP/VSVG DNA plasmids (Thermo Fisher Scientific). Lentiviruses were harvested 48 and 72 h post-transfection, pooled, filtered through 0.45 μm filter (Sartorius) and used to infect HeLa cells immediately. HeLa cells seeded in 24-well plate were infected twice every 24 h with filtered lentiviruses in the presence of 8 μg ml$^{-1}$ of hexadimethrine bromide (Sigma-Aldrich, #H9268). After incubation for another 24 h, cells were transfected with mCherry-P4M and processed for immunofluorescence after 24 h. The knockdown efficiency was quantified using RT-qPCR, the procedure of which is described below. The total RNA was extracted using TRIzol$^{TM}$ reagent (Thermo Fisher Scientific). After reverse transcription (Primerdesign), SYBR-green-based real-time PCR (Primerdesign) was performed in Bio-Rad CFX96 Touch$^{TM}$ real-time PCR detection system. Data were normalized by GAPDH. The primer pairs for PI4KIIIβ and GAPDH are (5′-GGA GAA TGA GGA TGA GGA GCT CTC C-3′ and 5′-GCC AGT CGA ACA GGG GAA CTG AAT G-3′) and (5′-GGA GTC CAC TGG CGT CTT-3′ and 5′-TCT TGA TGC TGT TGT CAT ACT T-3′), respectively.

**The production of antibodies**. BL21 DE3 E. coli bacteria were transformed with DNA plasmids encoding GFP-6 × His, Dopey1(1171–1320)-6 × His or Dopey2 (1005–1188)-6 × His, GST-GFP, GST-Dopey1(1171–1320), and GST-Dopey2 (1005–1188). Resulting bacteria were then induced by isopropyl β-D-1-thiogalactopyranoside (IPTG) and the bacteria were subsequently pelleted. His-tagged fusion proteins were purified by sonication in 8 M urea phosphate-buffered saline (PBS). The supernatant after centrifugation was incubated with Ni-NTA agarose beads for 2 h at room temperature and the bound protein was eluted in 200 mM imidazole urea. After dialysis in PBS, the denatured protein was used as an antigen and antisera from immunized rabbits were collected by Genemed Synthesis Inc. GST-tagged fusion proteins were purified by sonication in lysis buffer (50 mM Tris, pH 8.0, 100 mM NaCl, 0.1% Triton X-100 and 1 mM dithiothreitol). After centrifugation, the resulting supernatant was incubated with glutathione Sepharose beads (GE Healthcare) for 4–12 h at 4 °C cold room. After extensive washing with lysis buffer, GST-fusion protein was cross-linked onto the beads by incubation with dimethyl pimelimidate (Sigma-Aldrich) in 200 mM sodium borate solution pH 9.0. The cross-linker was neutralized by incubation with ethanolamine and the cross-linked beads were incubated with the respective antisera. The bound antibody was eluted using 100 mM glycine pH 2.8. To obtain the antigen-depleted control antibody, Dopey1 or Dopey2 anti-serum incubated with cross-linked corresponding antigen was used to purify the total IgG by Protein A beads (Thermo Fisher Scientific). The control IgG was purified from pre-immunized rabbit serum using Protein A beads. Protein A bead bound IgG was eluted using 100 mM glycine pH 2.8. All the eluted antibodies were dialyzed against a buffer consisting of 100 mM NaCl and 40 mM HEPES pH 7.3, concentrated and stored at −20 °C supplemented with 50% glycerol.

**Production of detergent-free GST fusion proteins**. To purify detergent-free GST-DEC, GST-DEC(LKRL-4A) and GST-P4M fusion proteins, transformed BL21 DE3 E. coli bacteria were induced by IPTG and the bacteria were subsequently pelleted and lysed by sonication in the lysis buffer (50 mM Tris, pH 8.0, 100 mM NaCl). After clearing the lysate by centrifugation, the supernatant was incubated with glutathione Sepharose beads (GE Healthcare) in a cold room overnight. Beads were washed by the lysis buffer followed by elution in the lysis buffer containing 10 mM reduced glutathione (Sigma-Aldrich). The eluted protein was dialyzed against a buffer consisting of 100 mM NaCl and 40 mM HEPES pH 7.3. The protein was then concentrated by ultrafiltration spin column (GE Healthcare), quantified by Coomassie gel staining and stored at −20 °C until use.

**Immunoprecipitation**. HEK293T cells were used for transfection to express various tagged proteins. Cells were subsequently lysed in the lysis buffer (40 mM HEPES pH 7.3, 100 mM NaCl, 0.1 or 1% Triton X-100). After 30 min rotation in a cold room, the lysate was cleared by centrifugation at $17,000 \times g$ for 30 min. Next, the resulting supernatant was incubated with 0.5–1 µg antibody for 2–12 h or 20 µl GFP-Trap agarose beads (ChromoTek) for 2 h in a cold room. After that, the antigen-antibody complex was retrieved using 20 µl Protein A/G beads (Thermo Fisher Scientific). After washing with the lysis buffer, beads were subjected to boiling in 50 µl 2× sodium dodecylsulphate (SDS) sample buffer to elute bound proteins. At last, Western blot was performed to analyze bound proteins according to standard protocol. SDS polyacrylamide gel electrophoresis (PAGE) separated proteins were transferred to polyvinyl difluoride membrane (Bio-Rad), which was sequentially incubated with primary and HRP-conjugated secondary antibody. Western blot and molecular weight marker bands were acquired under the chemiluminescence and white-light imaging mode, respectively, using a cooled charge-coupled device of LAS-4000 (GE Healthcare Life Sciences). Molecular weights were manually assigned by aligning the two images. Uncropped blot images are presented in Supplementary Fig. 9.

**Large-scale IP and proteomics**. HEK293T cells grown in 9 φ15 cm Petri-dishes were lysed in the lysis buffer (40 mM HEPES pH 7.3, 0.1% Triton X-100 and 120 mM NaCl) and centrifuged at 100,000 g for 60 min. The cell lysate was pre-cleared by Glutathione Sepharose beads (GE Healthcare) and incubated with either control IgG, Dopey1 or Mon2 antibody for 4 h at 4 °C. Protein A/G agarose beads were subsequently added to the cell lysate to capture the antibody. After extensive washing using the lysis buffer, bead-immobilized proteins were eluted by boiling in SDS-sample buffer and separated in SDS-PAGE. After Coomassie staining, gel slices were cut and sent for liquid chromatography–mass spectrometry[2] analysis.

**In vitro transcription and translation**. TNT™ T7 Quick Coupled Transcription and Translation System (Promega) was used to in vitro transcribe and translate DMyc-TPR, DMyc-Dopey1(894–1247), DMyc-Dopey1(1240–1586), DMyc-Dopey1(1570–1894), GFP-MEC and DMyc-MEC according to manufacturer's protocol. Briefly, the reaction mixture containing TNT™ Quick Master Mix, methionine (1 mM) and plasmid DNA was incubated at 30 °C for 90 min. After verifying the protein expression by immunoblot, proteins were subjected to IP or pull-down. IP was performed using 15 µl GFP-Trap agarose beads as described above; GST pull-down was performed by incubating 20 µg of GST-DEN or GST-DEN(W34A) on Glutathione beads with in vitro translated DMyc-TPR overnight.

After washing beads with the lysis buffer (40 mM HEPES pH 7.3, 100 mM NaCl, 0.1 or 1% Triton X-100), bound proteins were analyzed in Western blot.

**Liposome pull-down**. Chicken egg phosphatidylcholine (PC), porcine brain phosphatidylethanolamine (PE), porcine brain phosphatidylserine (PS), bovine liver PI, porcine brain PI4P, porcine brain PI(4,5)P₂, chicken egg PA, 18:1 PI(3)P, 18:1 PI(5)P, 18:1 DAG and 18:1 LPA were purchased from Avanti Polar Lipids Inc. The base liposome was a mixture of 69% PC, 14% PE and 17% PS (mass percentage or m/m, same for the rest).

To prepare liposomes containing phosphoinositide or PA-derivative, 10% PC of the base liposome was substituted by PI, PI3P, PI4P, PI5P, PI(4,5)P₂, PA, DAG, or LPA. Lipids were mixed in desired proportion in chloroform, which was then removed by vacuum, leaving behind a thin lipid film. The lipid film was rehydrated in a buffer (40 mM HEPES, 100 mM NaCl, pH 7.3) to obtain a final concentration of 1 mg ml⁻¹ total lipids. Liposomes were generated after bath sonication. For pull-down assays, liposomes were first mixed with GST-fusion proteins or cell lysates expressing protein(s) of interest at room temperature for 1 h. Next, liposomes were pelleted in a table top centrifuge, washed and dissolved in 20 µl 2× SDS-sample buffer. Last, the elute was subjected to SDS-PAGE and Western blot analysis to detect bound proteins. To determine the binding affinity of GST-DEC and -P4M, base liposomes containing 1% PI4P were incubated with increasing concentration of GST-fusion protein and liposome-bound proteins were quantified by Western blot.

**Immunofluorescence**. Cells grown on No. 1.5 glass coverslips were first fixed by 4% paraformaldehyde in PBS and subsequently neutralized by NH₄Cl. Next, cells were incubated with primary antibodies diluted in antibody dilution buffer, which is PBS supplemented with 5% fetal bovine serum, 2% bovine serum albumin, and 0.1% Saponin (Sigma-Aldrich). After PBS washing, cells were incubated with secondary antibodies diluted in antibody dilution buffer. At last, the coverslip was mounted in Mowiol 4–88 (EMD Millipore) after extensive washing using PBS.

**Imaging-based trafficking assays**. To assay the trafficking of artificially tethered peroxisomes, cells co-expressing PEX3-mRFP-FKBP and FRB-fused Dopey1 fragment were treated with 50 nM rapamycin. The system was subsequently either imaged live under the wide-field microscope or incubated for various lengths of time before being processed for immunofluorescence.

For the following transport assays, cells were first subjected to control, Mon2 or Dopey1 knockdown. To assay the ER-to-Golgi trafficking of VSVGtso45-GFP, cells were further transfected to express VSVGtso45-GFP. After incubation at 40 °C overnight, cells were transferred to 32 °C to chase for various lengths of time in the presence of 10 µg ml⁻¹ cycloheximide, fixed and processed for immunofluorescence labeling of GM130 as the Golgi marker. To assay the ER-to-Golgi and Golgi-to-PM trafficking, cells were transfected to express a RUSH reporter: Ii-Strep_TNFα-SBP-GFP, ss-Strep-KDEL_ss-SBP-GFP-E-cadherin or Ii-Strep_ManII-SBP-GFP[34]. Residual amount of biotin in the cell culture medium was quenched by adding 50 ng ml⁻¹ streptavidin. Totally, 15–20 h after the transfection of RUSH reporters, cells were thoroughly washed to remove streptavidin. For the ER-to-Golgi trafficking assay, 40 µM biotin and 10 µg ml⁻¹ cycloheximide were added to the cell medium to release RUSH reporters from the ER. The system was subsequently either imaged live under a wide-field or spinning disk confocal microscope or chased for various lengths of time before being processed for immunofluorescence. For the Golgi-to-PM trafficking assay, cells were first incubated at 20 °C for 3 h in the presence of 40 µM biotin and 10 µg ml⁻¹ cycloheximide to synchronize RUSH reporters at the Golgi. The system was subsequently warmed up to 37 °C and either imaged live or chased for various lengths of time before being processed for immunofluorescence.

To assay the PM-to-Golgi trafficking, cells were transfected to express one of the following CD8a-fused cytosolic tails of TGN resident membrane proteins (CD8a-TGN-tails): CD8a-furin, CD8a-CI-M6PR, or CD8a-sortilin. The selective labeling of surface-exposed pool of chimera was performed by first incubating with anti-CD8a antibody on ice for 1 h. After washing away unbound antibody, cells were warmed up to 37 °C in complete medium to allow the internalization of the antibody for various lengths of time before being processed for immunofluorescence.

**Microscopy**. Fixed cells were imaged under an Olympus IX83 inverted wide-field microscope system equipped with an oil 63×/NA 1.40 objective (Plan Apo), an oil 100×/NA 1.40 objective (Plan Apo), an oil 40×/NA 1.20 objective (fluorite), a motorized stage, a focus drift correction device, a 37 °C enclosed environment chamber, motorized filter cubes, a scientific complementary metal oxide semiconductor camera (Neo; Andor) and a 200 W metal-halide excitation light source (Lumen Pro 200; Prior Scientific). Dichroic mirrors and filters were optimized for GFP/Alexa Fluor 488, mCherry/Alexa Fluor 594 and Alexa Fluor 647. The system was controlled by MetaMorph (Molecular Devices).

Live cell imaging was conducted under either a wide-field (by default) or spinning disk confocal microscope. The spinning disk confocal system comprised an Olympus IX81 inverted microscope equipped with an oil 100×/NA1.40 objective (Plan Apo), a piezo-motorized stage, a 37 °C enclosed environment chamber, 488 and 561 nm diode lasers, an electron-multiplying charge-coupled device

(C9100-13; Hamamatsu) and CSU-X1 spinning disk scan head (Yokogawa). Dichroic mirrors and filters are optimized for GFP and mCherry imaging. The whole microscope system was controlled by Volocity (PerkinElmer). For live cell imaging, cells were grown on the glass-bottom Petri-dish (MatTek Corporation) and subjected to knockdown or expression of FP-tagged proteins. During imaging, cell medium was replaced by $CO_2$-independent medium (Thermo Fisher Scientific) supplemented with 10% fetal bovine serum and 4 mM glutamine.

FRAP experiments were acquired by an inverted Nikon Eclipse Ti microscope equipped with an oil 100×/NA 1.4 objective (Plan Apo), a focus drift correction device, a piezo-motorized stage, 37 °C heated chamber (Lab-Tek), 100 mW diode lasers (491 and 561 nm), an electron-multiplying charge-coupled device (Evolve 512; Photometrics), CSU-22 spinning disk scan head (Yokogawa) and a 3D FRAP system (iLAS;[2] Roper Scientific). The microscope system was controlled by MetaMorph and iLAS[2] software (Roper Scientific). During FRAP experiments, 2D time lapse images of cells expressing GFP-ERGIC53 were collected before and after the photo-bleaching. The photo-bleaching region was manually drawn to include an area within the cell boundary but outside the Golgi, which was identified using co-expressed GalT-mCherry signal.

**Stoichiometry by single-molecule photo-bleaching steps.** No. 1.5 Φ25 mm glass coverslips were cleaned by alternate NaOH and ethanol treatment in a bath sonicator. After extensive rinse in water, coverslips were dried in an oven. HEK293T cells were transiently transfected to express GFP-Dopey1, GFP-Mon2 or GFP-Dopey1, and DMyc-Mon2. Cells were subsequently lysed in a hypotonic buffer (20 mM Tris pH 7.4) by passing them through 30 G gauge needle 20 times on ice. The resulting cell lysate was spun at high speed in a refrigerated table-top centrifuge and the supernatant was applied onto a precleaned coverslip for 1 h at room temperature in the dark. After washing with water, the coverslip was dried in a vacuum and assembled into a coverslip chamber with lysate side up. Next, a 15% acrylamide/bis-acrylamide gel in a buffer (20 mM Tris pH 7.4, 50 mM NaCl) was overlaid on the glass coverslip in the chamber. After the gel was set, GFP single-molecules were imaged under a TIRF microscope comprising Zeiss Observer Z1 inverted microscope equipped with a 100×/NA 1.46 objective lens (plan apochromat), a fully motorized stage, Zeiss Laser TIRF 3 module, 30 mW solid state laser (488 nm), Zeiss Filter Set 52 HE (488 nm and shift free) (excitation filter: BP 488/20, dichroic mirror: FT505 and emission filter 530/550) and an electron-multiplying charge-coupled device (Evolve 512). The microscope system was controlled under Zen software (Zeiss). Time lapse movies were captured with 20 ms exposure time at a frequency of ~30 Hz. Image analysis was performed using ImageJ (http://imagej.nih.gov/ij/). The maximum intensity projection image of the first 10 frames was acquired and all fluorescence objects within the image field were manually selected using round regions of interests (ROIs) with a diameter of 7 pixels. The intensity vs time profile of each object was subsequently analyzed in Excel (Microsoft). The single-molecule photo-bleaching event was visually defined as a single-frame large intensity drop occurring between frame 10 to $m - 11$ ($m$ is the number of frames of the time lapse and the first frame is frame 0). For an intensity trace, we were able to unambiguously identify at least three single-molecule photo-bleaching events. Analyzable fluorescence objects (~40% of total objects) were those completely photo-bleached before frame $m - 11$ through 1–3 steps of single-molecule photo-bleaching events. Three independent experiments were conducted and each considered ≥100 analyzable objects.

**Image analysis.** All image analysis was conducted by using ImageJ. Data analysis and plotting were conducted in OriginPro 9 (OriginLab).

For Golgi intensity measurements, the ROI of the Golgi was generated by intensity segmentation using the co-stained Golgi marker. The image was background-subtracted by using ROIs outside cells. In Fig. 3g, h and 4d, each independent experiment analyzed ≥279, 168, and 133 cells, respectively; in Supplementary Fig. 4c, 4e, and 6o, each independent experiment analyzed ≥133, 13, and 50 cells, respectively.

The below method was used to analyze the Golgi fraction. In the channel of the reporter fluorescence, $A_{cell}$ and $A_{Golgi}$ are the area (in pixels) of the cell and the Golgi ROI respectively, while $I_{cell}$ and $I_{Golgi}$ are the mean intensity of the cell and the Golgi ROI, respectively. $f$ is a constant value between 0 and 1. $f = 0.5$ was used for our calculation. The fraction of Golgi-localized reporter was calculated as ($I_{Golgi} - f^*I_{cell})^*A_{Golgi}/((1 - f)^*I_{cell}^*A_{cell})$. In each image to be quantified, all cells positively expressing the reporter of interest were analyzed.

For FRAP, the Golgi ROI was generated by intensity segmentation of GalT-mCherry and the cell ROI was manually drawn. After photo-bleaching, the integrated intensity of GFP-ERGIC53 within the Golgi ROI ($I_{Golgi}$) and the cell ($I_{cell}$) were calculated after background subtraction. The ER fraction of GFP-ERGIC53 at the post-bleaching time $t$, $F_t$, was calculated as ($I_{cell} - I_{Golgi})/I_{cell}$. A series of $F$ values were fitted to a single exponential decay function $y = y0 + A1^*exp (-(x - x0)/t1)$ in OriginPro 9 to obtain $t_{1/2}$ ($t_{1/2} = 0.693t1$), adjusted-$R^2$ and $y0$. Only curves with adjusted-$R^2 \geq 0.8$ were selected for further analysis. $y_{x=0}$ was calculated as $y_{x=0} = y0 + A1^*exp(x0/t1)$. ($F_t - y_{x=0})/(y0 - y_{x=0})$ was subsequently plotted against time ($t$) to obtain a normalized plot.

Normalized gyradius was calculated to quantify the ensemble distribution of an

intracellular organelle. A cell of interest was first extracted as a new image. The image was subsequently background-subtracted so that intensities of background pixels were zero. Next, the coordinate of the center of fluorescence mass was acquired by ImageJ (Analyze > Set Measurements and Analyze > Measure). Assuming that $I_i$ is the intensity of pixel i and $r_i$ is the distance of pixel i to the center of fluorescence mass, the gyradius of the organelle, $R_{organelle}$, was calculated as

$$R_{organelle} = \sqrt{\frac{\sum (I_i \cdot r_i^2)}{\sum I_i}}, \tag{1}$$

with all pixels included. The calculation was automated by an in-house compiled ImageJ Macro. A binary image tracing the cell contour was manually drawn from the original image so that pixel values of the cell contour were 1 while the rest were 0. The gyradius of the cell contour, $R_{cell}$, was similarly calculated. The normalized gyradius was defined as $R_{organelle}/R_{cell}$, which was used to quantitatively compare organelle distribution between the cell center and periphery.

**Particle analysis.** For particle analysis, 2D time lapse images were acquired at 3–5 Hz for a duration of 11–22 s under a wide-field microscope. Particles were manually tracked using mTrackJ[61] plugin of ImageJ to acquire coordinates of particles ($x$, $y$) at each frame or time point. We adopted a polar coordinate system to describe particle's radial movement between the cell center and periphery. Since particles are moving along microtubule arrays, which are radiating from the cell center to periphery, only radial component of the movement is considered and the centrifugal direction was defined as positive. For a cell of interest, the reference point, ($x_0$, $y_0$), was defined as the center of its fluorescence mass at the first frame of the time lapse. A particle's position can be described by its distance to the reference point without considering its angle component, therefore greatly simplifying the quantitative analysis. A track is defined as the trajectory of a particle from the beginning to the end of a 2D time lapse image. Only particles that form tracks were analyzed. Radial instantaneous velocity or velocity (μm s$^{-1}$) is the radial displacement of a particle between two consecutive frames divided by the time interval between frames. The single particle mean velocity was calculated as $\frac{\sum_{j=1}^{m-1} v_j}{m-1}$, in which, $v_j$ is the velocity of the particle at frame j; m is the number of frames of the time lapse (the first frame is frame 0). For the ensemble particle analysis, the mean of single particle mean velocities was calculated as $\frac{\sum_{i=1}^{n} \sum_{j=1}^{m-1} v_{i,j}}{n \cdot (m-1)}$, in which, $v_{i,j}$ is the instantaneous velocity of a particle i at frame j; m is the number of frames (the first frame is frame 0); and $n$ is the number of particles or tracks. The mean of single particle mean velocities represents the ensemble net movement of particles.

A particle in a track can move continuously, pause or reverse its direction of movement. Within a track, a segment is defined as a period of uninterrupted and unidirectional movement (≥3 consecutive frames), in which a particle's instantaneous velocity always either ≥0.1 (centrifugal segment) or ≤ −0.1 μm s$^{-1}$ (centripetal segment). Segment velocity (μm s$^{-1}$; scalar: segment speed) is the mean instantaneous velocity within a segment and was calculated as the sum of all instantaneous velocities divided by the number of frames within the segment. Segment time (s) is the duration of a segment and was calculated as the number of frames of the segment multiplied by the frame interval of the time lapse images. Segment frequency (Hz) of a track is the number of all centrifugal or centripetal segments within the track divided by the total time span of the track (or the time lapse movie). Segment displacement (μm; scalar: segment distance) is the radial displacement that the particle travels within the segment. Centrifugal bias of segment speed (μm s$^{-1}$), centrifugal bias of segment time (s), centrifugal bias of segment frequency (Hz), and centrifugal bias of segment distance (μm) were calculated as the centrifugal segment parameters minus the corresponding centripetal ones.

In the ER-to-Golgi trafficking assays, cargos appear as puncta which can be the ERES, ERGIC or membrane carriers. In particle analysis, we excluded particles which traveled ≤400 nm in distance from the start to the end of the time lapse (17.15 s). ERESs/ERGICs and membrane carriers were not further distinguished for our subsequent particle analysis. Those excluded particles are mainly ERESs/ERGICs, which are known to be largely immobile, and carriers, which happen to pause during the time lapse imaging.

**Deconvolution.** Wide-field image stacks (z step size: 150 nm) acquired for the ER-to-Golgi trafficking assay were deconvolved using Classic Maximum Likelihood Estimation method of Huygens Professional software package (Scientific Volume Imaging).

## Data availability

Data supporting the findings of this manuscript are available from the corresponding author upon reasonable request. A reporting summary for this Article is available as a Supplementary Information file. The source data underlying Figs. 1g, 2d, e, 3b, 3d, 3f–h, 4d, 5g, 5j–p, 6d, 6f, 7b, c, 7e–l, 8b, c, 8f–l and Supplementary Figs. 1m, 2d, 2f, 3d, 3f, g, 3j, 3l, 3n, o, 3q, 3s, 4c, 4e, 5g, 5i, 5k, 5m, 6k, 6m, 6o, 7b-c, 7g, h, 8d are provided as a Source Data file.

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

## Acknowledgements

We would like to thank W. Hong (Institute of Molecular and Cell Biology, Singapore) for the initial support of this work and B. Antony (CNRS, France), T. Balla (National Institutes of Health, USA), H. Hauri (University of Basel, Switzerland), R. Irvine (University of Cambridge, UK), T. Kirchhausen (Harvard Medical School, USA), J. Lippincott-Schwartz (National Institutes of Health, USA), F. Perez (Institut Curie, France), and M. Vaughan (National Institutes of Health, USA) for sharing DNA plasmids. This work was supported by the following grants to L.L.: MOE AcRF Tier1 RG132/15, Tier1 RG35/17, and Tier2 MOE2015-T2-2-073.

## Author contributions

L.L. conceived and supervised the study. L.L and D.M. designed the experiments. D.M. performed a majority of experiments. D.M. and L.L. analyzed the data. H.C.T cloned human Dopey1. B.C. generated polyclonal antibody against Dopey1 and Dopey2. L.L. and D.M. prepared figures and wrote the paper.

## Additional information

**Competing interests:** The authors declare no competing interests.

