## [Transparent Peer Review File · Nature Communications]

Reviewers' comments:

Reviewer #1 (Remarks to the Author):

This is very interesting manuscript presenting an overwhelming amount of data conveying the message that the Mon2-Dopey1 complex engages kinesin1 and exerts an important role in organelle positioning and in the outward movement of membrane-bound carriers.

Overall, the data (with limited exceptions, see below) sustain the conclusions drawn by the authors, yet there are some aspects that should be clarified.

1. The involvement of PI4P as a factor required for the Golgi localization of Dopey1 is investigated through the use of PAO, a rather non-specific inhibitor of PI4Ks, that also inhibit tyrosine kinases and other kinases. The involvement of PI4P should be explored using more specific tools, including more specific inhibitors of PI4Ks and via the siRNA-mediated depletion of the different PI4K isoforms.
2. Similarly, the use of primary alcohol to inhibit PLD should be corroborated by more specific approaches: more specific PLD inhibitors and depletion of PLDs via siRNA.
3. The liposome-binding assay using cell lysates gives a very low signal-to-noise ratio as for GFP-DEC domain (Fig. 3D shows a high background signal using lysates from cells transfected with GFP alone). Also the signal in panel 3G for GFP-dopey1 is very weak and not much higher than the hypothetical negative control (3H). an between and GFP dopey1 MEANING? More stringent assay conditions should be applied to ascertain the specificity of the pull-down of Dopey1.
4. The interaction of Mon2 with PA deduced from liposome-binding assays with cell lysates may be indirect: no formal proof of a direct interaction is provided in the manuscript and a binding assay with isolated recombinant proteins is needed to draw this conclusion.
5. The interaction of the Mon2-Dopey1 complex with kinesin is also observed by co-IP studies or liposome pull-down assays from lysates of cells overexpressing the relevant proteins/domain and do not allow to conclude that the interaction is direct. Indeed, the "disproportionate" amount of kinesin that is pulled down in the liposome assay from lysates of cells co-expressing GFP-Dopey1, mCherry-Mon2 and Flag-KLC2 (4E) is not consistent with a stoichiometric ratio between KLC2 itself and Dopey1. This casts doubt on the conclusion that it is Dopey1 that is mediating the pulldown of KLC2. Also in this case, to draw a definitive conclusion about a direct interaction between Dopey1 and KLC2, an interaction between recombinant proteins should be documented.
6. The authors propose that Mon2-Dopey1 might function as a quasi-universal adaptor for kinesin-1 on a variety of organelles: the Golgi complex, TGN-derived carriers, carriers travelling between the ER and the Golgi, recycling endosomes, late endosomes, endosomes or endosome-derived carriers directed to the Golgi complex and lysosomes. What is not clear in this scenario is how, with the exception of the TGN and TGN-derived membranes, the membrane-targeting of Mon2 to so many different organelles could occur, considering the very low affinity for PI4P (which is abundant at the Golgi and present but not highly concentrated elsewhere) and above all considering that Mon2 membrane association, at least with the Golgi complex, is sensitive to BFA, suggesting that it involves ARF1 or ARF1-dependent effectors. Is the association of Mon2-Dopey1 with non-Golgi sites (that is, however, only documented with overexpressed proteins in the manuscript) also PI4P-dependent and BFA-sensitive?

Reviewer #2 (Remarks to the Author):

In this manuscript, the authors investigate the role of a novel Dopey1-Mon2 complex in membrane

trafficking and the organisation of the endomembrane system. Following a proteomic analysis for Mon2 interactors that implicated Dopey2, the authors show by immunoprecipitation that endogenous Mon2 interacts with Dopey1 but not Dopey2. They map key regions involved in the interaction in both proteins: The 'MEC domain' of Mon2 and the central region of Dopey1. In addition, Mon2 appears to self-associate to promote the formation of at least a tetrameric complex. The complex localises to the Golgi apparatus and the localisation of Dopey1 is dependent on Mon2. The authors go on to identify lipid binding determinants in both proteins which appear to support this Golgi localisation; with Dopey1 interacting with PI4P and Mon1 with PA. Following another mass-spec based screen for binding partners, the authors provide data suggesting that Dopey1 interacts with the kinesin-1 microtubule motor via its light chain (KLC2) subunit and biochemically implicate a tryptophan-alanine motif within the amino terminal domain of Dopey1. A capacity to recruit kinesin-1 is further substantiated by assays that suggest that forced Dopey1 recruitment to peroxisomes promotes their centrifugal transport. The authors go on to investigate a role for the Mon2-Dopey1 complex in organelle positioning and conclude that they are important for spatial organisation of the RE, LE, lysosome and ERES/ERGIC compartments and a variety of different membrane trafficking events to and from the Golgi. If so, this would be an important advance.

Although the work appears to be generally performed well (in particular the detailed biochemical experiments in the early figures describing the complex, its key protein-protein and protein-lipid interactions and their effects on localisation are quite strong), I find myself, as yet, not convinced by the overall conclusions of the study. The authors go to great effort to define apparently very specific Golgi localisation determinants (at least for the endogenous proteins), but then when it comes to functional characterisation appear to be suggesting that this complex is important for the generic kinesin-1 dependent transport of a very wide range of organelles, even though the complex doesn't obviously localise to them at its endogenous expression level. It seems much more likely to me that the consequences for other elements of the endomembrane system are an indirect effect resulting from a perturbation of the Golgi function of this complex. Moreover, it is surprising not to see the various mutations defined in the early part of the manuscript not incorporated at all into the later functional analysis – for example, determining the effect of the LRRK/4A and W/A mutations on the trafficking/positioning phenotypes.

Specific Comments:

1. I am not convinced that the data presented in Figure 1 adequately support the very explicitly stated stoichiometry of the complex. Whilst the data are indeed consistent with at least a dimeric Mon2 that associates with 2x Dopey1, a higher order assembly also seems possible. Unless, more direct evidence can be provided (perhaps using purified recombinant proteins and appropriate biophysical analysis), the authors should tone down this conclusion. Regardless, it would seem relatively straightforward to formally demonstrate the direct domain-domain interactions within the complex and with KLC that are inferred from the co-IP experiments.

2. The authors implicate a tryptophan residue in the DEN domain as important for KLC2 binding. The various western blot data supporting this are convincing, but the authors mechanistically connect this previously reported WD motifs found in some other kinesin-1 adaptors, despite some sequence differences. Moreover, the WA motif appears to be found within a predicted alpha helical region (LZ?) of Dopey1, whereas validated WD motifs have so far been found in predicted unstructured regions of proteins. The authors suggest in their schematic that this region is a potential leucine zipper. They might consider the extensive literature on JIP3/4-KLC binding which appears to occur via a LZ-TPR interaction in their narrative. Regardless, the authors should incorporate this mutation into the functional analysis of peroxisome relocalization, organelle positioning and Golgi-trafficking to demonstrate the importance of the link to kinesin-1 for its function.

3. Similarly, the authors should determine the effect of lipid binding mutations in their

positioning/trafficking functional assays.

4. Despite the quantification, I would like to see representative fields of cells, not just single cells when describing the organelle positioning phenotypes as this can vary considerably from cell to cell in a single experiment and the extent of these changes is not readily apparent from simply considering the gyradius measurement provided. In addition, I cant see a description of this quantitative approach in the methods as the main text indicates.

5. The authors use the SUK4 antibody to detect Kif5B. They should confirm the specificity of this antibody in their IF experiments using siRNA knockdown.

6. It is not clear why the proteomic analysis that informs these studies is not described in the supplementary data.

7. Figure1A - please show directly comparable regions of the reprobed upper and lower panels i.e. the region of the Mon2 blot that corresponds to the location of the Dopey2 band the Dopey2 reprobe.

8. What effect does Mon2 knockdown have on Dopey2 Golgi-localisation. Is this consistent with the binding data in figure 1?

9. In general, molecular weight maker annotations should be provided for the western blots.

10. The authors state that low pH buffer 'activates kinesin-1' based on work by Heuser. This overstates current knowledge – for still unclear reasons, this effect is quite specific to lysosomes and late endosomes and may or may not be caused by a direct activation of kinesin-1 or could regulate upstream, lysosome specific pathways.

11. Figure 3 – the effect of PAO on Dopey1 Golgi localisation should be quantified. There also appears to be some reduction in Golgi 245 staining that may confound the interpretation.

12. For the liposome co-sed experiments, how do the authors know that all liposomes formed and pelleted with similar efficiency, which would be important for their interpretation of binding specificity?

Reviewer #3 (Remarks to the Author):

The authors present a very interesting and comprehensive analysis of a novel kinesin-1 adaptor complex, the Dopey1-Mon2 complex. What is particularly engaging is the thorough characterisation of the lipid-mediated recruitment (which is very novel in the kinesin field) and the highly quantitative characterisation of organelle distribution changes in response to perturbations. This work will be of broad interest to the cell biology community and I recommend publication with minor revisions. My comments are outlined below in rough order of priority.

1) In general there is not enough description of how analyses were performed (e.g. golgi localisation quantification), and in particular which statistical tests were performed to generate the p values. A description of analysis methodology is needed and which tests were used in the figure legends. I realise that single cell analysis has been carried out for the most part, but there is also no clear indication of how many biological replicates were done for each experiment (presumably at least three?). In general the manuscript comes across as comprehensive and well considered so I have concluded that this omission was by accident rather than design - although it is obviously difficult to review statistical rigor without these details. These omissions are throughout the

manuscript in all figures.

2) In the methods it is apparent that both HeLa cells and HEK cells are used, but it is unclear which cell lines are used for which experiment when reading the manuscript or referring to the figure legends.

3) In general the authors overstate their confidence in the stoichiometry of the complex and the 'directness' of interactions given the experiments they have done. Although I agree that Mon2 and Dopey1 likely multimerise, statements about dimerisation and heterotetramerisation - particularly in the first part of the manuscript (lines 58 to 82, including the subtitle) - can't really be justified. Later on with the artificial dimerisation it does suggest this for Dopey, but I would not conflate this with the earlier results. Similarly, in line 211: replace "directly interacts" with complexes. Avoid the speculation of bivalent interactions at line 218 (it's somewhat under debate how many WD motifs are needed to activate kinesin through KLC2).

4) The authors maintain that the DEC domain is not conserved between Dopey1 and Dopey2 (line 106), however a quick look at the alignment between these two shows that the C-terminal sequences are broadly very similar. In fact the LKRL motif and flanking residues of Dopey1 are almost identical in Dopey2, but the motif carries a change to LKRQ. I would say the striking difference between Dopey1 and Dopey2 is even more interesting in this light, rather than trying to say the C-terminus of Dopey1 is entirely unique (line 107). Please adjust the way this section is phrased to better reflect the sequence identity between Dopey1 and Dopey2.

5) In general I find the author's use of "DEC", "DEN", "MEC" etc to be quite confusing and a little meaningless. There are no such domains, just protein fragments. In the case of "DEC" it seems that it is not the DEC domain that is necessary for targeting to the Golgi, but the LKRL motif. Similarly it is not the DEN domain that binds to KLC2, but the WD motif. We can agree to disagree, I only ask that the authors consider whether it really adds to the clarity of the paper.

6) Supplemental Figure 2C,D (line 88 in manuscript), the siRNA knockdown and stability isn't quantified so it is difficult to gauge how reproducible the knockdown was.

7) Lines 100-101: you mention a highly conserved LKRL motif from analysing sequences, could you include a cross species alignment to verify this for the reader?

8) Line 105: Please show images that reflect the cytosolic distribution of Dopey2 C-terminal fragments as claimed.

9) Please quantify Figure 3A-C in a similar manner to Figure 2D/E, but showing the changes over time. Similarly Supplemental Figure 3I.

10) Has Supplemental Figure 3J,K been quantified? The P14P signal looks less in the Mon2 siRNA condition.

11) Is the multiple vesicular organelle (e.g. EE, LE, lysosome etc etc) localisation of FP-tagged Mon2 and Dopey1 (Fig 5A and Sup Fig 5A) an overexpression artifact - do you see this localisation using endogenous antibodies?

12) Line 171: Did the ACC1-dimer also dissociate?

13) Supplemental Figure 3 has a leftover placeholder box in the bottom right hand corner.

14) Make it clear in the text (line 192) that Figure 4A is endogenous and SF4A is overexpressed.

15) Is there a graph for Figure 4E? How many repeats were done.

Reviewers' comments:

Reviewer #1 (Remarks to the Author):

This is very interesting manuscript presenting an overwhelming amount of data conveying the message that the Mon2-Dopey1 complex engages kinesin1 and exerts an important role in organelle positioning and in the outward movement of membrane-bound carriers.

Overall, the data (with limited exceptions, see below) sustain the conclusions drawn by the authors, yet there are some aspects that should be clarified.

1. The involvement of PI4P as a factor required for the Golgi localization of Dopey1 is investigated through the use of PAO, a rather non-specific inhibitor of PI4Ks, that also inhibit tyrosine kinases and other kinases. The involvement of PI4P should be explored using more specific tools, including more specific inhibitors of PI4Ks and via the siRNA-mediated depletion of the different PI4K isoforms.

Reply:

We have performed the suggested experiment by knocking down the endogenous PI4KIII β , the PI4K that is responsible for the production of the majority of Golgi PI4P (Godi et al., Nature Cell Biol., 1999). Two shRNAs were designed and demonstrated to work efficiently in knockdown by RT-qPCR. When the transcript of PI4KIII β was depleted by ~75% (quantified by RT-qPCR), we observed that the Golgi localization of endogenous Dopey1, but not Mon2, reduced significantly. New figures are added as Fig. 3g,h and Supplementary Fig. 3g,h.

2. Similarly, the use of primary alcohol to inhibit PLD should be corroborated by more specific approaches: more specific PLD inhibitors and depletion of PLDs via siRNA.

Reply:

Our PA-liposome pull-down demonstrated the interaction between PA and Mon2 (directly or indirectly) (Fig. 4d,e in the revamped manuscript) and suggested that PA might contribute to the membrane recruitment of Mon2. Since primary alcohol can inhibit PLD-catalyzed transphosphatidylation, we have performed the knockdown of PLD1 and PLD2 singly or doubly. Although we can achieve efficient knockdown of the transcripts of PLD1/2, we did not notice the change of the Golgi localization of endogenous Mon2 (data not shown).

Besides the possibility of incomplete depletion of PLD1/2, we provide another explanation for why the primary alcohol, but not the knockdown of PLD1/2, worked for the membrane dissociation of endogenous Mon2. In addition to PLD1/2, it is known that multiple lipid metabolic pathways produce and consume PA. Endogenous PA level might not change significantly after the slow depletion of PLD1/2 due to the gradual compensatory changes in other PA-metabolic pathways. However, acute (within minutes) inhibition of the production of PA by primary alcohol can reduce the cellular level of PA.

3. The liposome-binding assay using cell lysates gives a very low signal-to-noise ratio as for GFP-DEC domain (Fig. 3D shows a high background signal using lysates from cells transfected with GFP alone). Also the signal in panel 3G for GFP-dopey1 is very weak and not much higher than the hypothetical negative control (3H). an between and GFP dopey1 MEANING? More stringent assay conditions should be applied to ascertain the specificity of the pull-down of Dopey1.

Reply:

Figure 3D, G and H have now become Fig. 3i, l and m in the revamped manuscript.

We think that it probably makes more sense to compare intensities of bands within the same gel blot. In the gel blot shown in Fig. 3i, PI4P pulled down much more GFP-tdDEC, but not GFP, than other phosphoinositides. In the gel blot shown in Fig. 3l, GFP-Dopey1(1900-2456) pulled down by PI4P is much stronger than other bands within the same blot; in contrast, in Fig. 3m, LKRL-4A mutant pulled down by PI4P is of similar intensity as neighbour bands.

4. The interaction of Mon2 with PA deduced from liposome-binding assays with cell lysates may be indirect: no formal proof of a direct interaction is provided in the manuscript and a binding assay with isolated recombinant proteins is needed to draw this conclusion.

Reply:

Although the PA-liposome clearly pulled-down Mon2 expressed in the cell lysate (Fig. 4d,e in the revamped manuscript), we agree with this reviewer that our data did not indicate a **direct** interaction between PA and Mon2. At the moment, we are unable to prepare a purified fragment of Mon2 encompassing 1-1200 aa or longer to test the direct interaction. We have modified the statement that the PA-Mon2 interaction is direct.

5. The interaction of the Mon2-Dopey1 complex with kinesin is also observed by co-IP studies or liposome pull-down assays from lysates of cells overexpressing the relevant proteins/domain and do not allow to conclude that the interaction is direct. Indeed, the “disproportionate” amount of kinesin that is pulled down in the liposome assay from lysates of cells co-expressing GFP-Dopey1, mCherry-Mon2 and Flag-KLC2 (4E) is not consistent with a stoichiometric ratio between KLC2 itself and Dopey1. This casts doubt on the conclusion that it is Dopey1 that is mediating the pulldown of KLC2.

Also in this case, to draw a definitive conclusion about a direct interaction between Dopey1 and KLC2, an interaction between recombinant proteins should be documented.

Reply:

1 Regarding the issue of the direct interaction:

Using liposomes and purified recombinant GST-DEC, the direct interaction between PI4P and DEC was shown in Fig. 3I (old version) or Fig. 3n (revamped version). We now provide further data to conclude that the following three interactions are direct too: MEC and MEC, MEC and Dopey1, and DEN and TPR. These experiments were conducted using in vitro translated and purified proteins. Unlike the mammalian cell lysate, the rabbit reticulocyte lysate used in the in vitro translation comprises almost solely the protein synthesis machinery. We demonstrated that in vitro translated MEC co-IPed itself (Fig. 1h, revamped version); that in vitro translated MEC co-IPed a fragment of Dopey1 comprising residues 894-1247 (Fig. 1e, revamped version); and that bead-immobilized GST-DEN pulled down in vitro translated TPR (Supplementary Fig. 5d, revamped version). The only interaction that we can't conclude as a direct one is the interaction between Mon2 and PA, as discussed in our reply to point 4 of Reviewer #1's comment.

2 Regarding the disproportionate amount of kinesin pulled down in Fig. 4E (old version; Fig. 5e in revamped version):

Not all Dopey1 molecules recruited to the liposome by PI4P are active and only a small fraction can probably engage KLC2. Therefore, the stoichiometric ratio between Dopey1 and KLC2 on the PI4P-liposomal membrane is expected to be much more than 1. The disproportionate amount of KLC2 pulled down (13.9-fold increase) is consistent with our model in which Dopey1 can be activated by Mon2, likely by dimerization. Comparing the 8th and 10th lanes that show the positive pull-down of both GFP-Dopey1 and Flag-KLC2, PI4P-liposomes recruit similar amount of Dopey1 (1.9-fold difference). However, the liposome of the 10th lane contains much more active Dopey1 to recruit KLC2, due to the dimerization of Dopey1 induced or activated by the co-expressed Mon2. The same trend has been repeatedly observed (see point 15 of reviewer #3 comment).

6. The authors propose that Mon2-Dopey1 might function as a quasi-universal adaptor for kinesin-1 on a variety of organelles: the Golgi complex, TGN-derived carriers, carriers travelling between the ER and the Golgi, recycling endosomes, late endosomes, endosomes or endosome-derived carriers directed to the Golgi complex and lysosomes. What is not clear in this scenario is how, with the exception of the TGN and TGN-derived membranes, the membrane-targeting of Mon2 to so many different organelles could occur, considering the very low affinity for PI4P (which is abundant at the Golgi and present but not highly concentrated elsewhere) and above all considering that Mon2 membrane association, at least with the Golgi complex, is sensitive to BFA, suggesting that it involves ARF1 or ARF1-dependent effectors. Is the association of Mon2-Dopey1 with non-Golgi sites (that is, however, only documented with overexpressed proteins in the manuscript) also PI4P-dependent and BFA-sensitive?

Reply:

As this reviewer pointed out, Mon2's membrane association should require Arf1 or its effectors in addition to the PA. It has been documented that Arf1 localizes to and functions in diverse secretory and endocytic membrane trafficking pathways. Both PA and PI4P are known to localize to the ERES/ERGIC and endolysosome in addition to the Golgi (see manuscript Discussion). Therefore, the localization of Dopey1 and Mon2 at multiple organelles and carriers is possible. However, it is likely that concentrations of Dopey1 and Mon2 at these non-Golgi organelles and carriers can vary greatly and be low.

We have tested if Dopey1 and Mon2's punctate localization is sensitive to PI4P depletion (induced by PAO) and Arf1 inactivation (induced by brefeldin A treatment) (Supplementary Fig. 6d,e in revamped manuscript). Under live-cell imaging, brefeldin A treatment clearly dissociates both GFP-tagged Mon2 and Dopey1 from the Golgi and puncta within 360 s; in contrast, only GFP-Dopey1 is affected under

600 s PAO treatment. Our data therefore are consistent with the model that Mon2's localization is dependent on Arf1 and Dopey1's localization requires both Mon2 and PI4P.

Reviewer #2 (Remarks to the Author):

In this manuscript, the authors investigate the role of a novel Dopey1-Mon2 complex in membrane trafficking and the organisation of the endomembrane system. Following a proteomic analysis for Mon2 interactors that implicated Dopey2, the authors show by immunoprecipitation that endogenous Mon2 interacts with Dopey1 but not Dopey2. They map key regions involved in the interaction in both proteins: The 'MEC domain' of Mon2 and the central region of Dopey1. In addition, Mon2 appears to self-associate to promote the formation of at least a tetrameric complex. The complex localises to the Golgi apparatus and the localisation of Dopey1 is dependent on Mon2. The authors go on to identify lipid binding determinants in both proteins which appear to support this Golgi localisation; with Dopey1 interacting with PI4P and Mon2 with PA. Following another mass-spec based screen for binding partners, the authors provide data suggesting that Dopey1 interacts with the kinesin-1 microtubule motor via its light chain (KLC2) subunit and biochemically implicate a tryptophan-alanine motif within the amino terminal domain of Dopey1. A capacity to recruit kinesin-1 is further substantiated by assays that suggest that forced Dopey1 recruitment to peroxisomes promotes their centrifugal transport. The authors go on to investigate a role for the Mon2-Dopey1 complex in organelle positioning and conclude that they are important for spatial organisation of the RE, LE, lysosome and ERES/ERGIC compartments and a variety of different membrane trafficking events to and from the Golgi. If so, this would be an important advance.

Although the work appears to be generally performed well (in particular the detailed biochemical experiments in the early figures describing the complex, its key protein-protein and protein-lipid interactions and their effects on localisation are quite strong), I find myself, as yet, not convinced by the overall conclusions of the study. The authors go to great effort to define apparently very specific Golgi localisation determinants (at least for the endogenous proteins), but then when it comes to functional characterisation appear to be suggesting that this complex is important for the generic kinesin-1 dependent transport of a very wide range of organelles, even though the complex doesn't obviously localise to them at its endogenous expression level. It seems much more likely to me that the consequences for other elements of the endomembrane system are an indirect effect resulting from a perturbation of the Golgi function of this complex.

Reply:

It is easier to study the Golgi localization since Dopey1 and Mon2 have a relatively strong presence at the Golgi. The determinants for Golgi localization of Dopey1 and Mon2 should also apply to their localization to other organelles, such as the ERES/ERGIC and endolysosome. We have evidence that Dopey1 and Mon2's Golgi and punctate localization have the same sensitivity toward PI4P depletion (induced by PAO) and Arf1 (induced by brefeldin A treatment) (Supplementary Fig. 6d,e in revamped manuscript), therefore suggesting the non-Golgi localization mechanism might be the same as that of Golgi localization. A small amount of endogenous Mon2 can also be detected at non-Golgi sites, such as the RE and lysosome (Supplementary Fig. 6b in revamped manuscript).

Sometimes, the dominant cellular localization does not always correlate with where a protein functions. A good example is kinesin-1 heavy chain, kif5b, which is well known for the motility of most non-Golgi organelles, including the mitochondrion, ERES, ERGIC and endolysosome. However, a predominant amount of Kif5b localizes to the Golgi, as revealed by SUK4 antibody (Marks et al., JCS, 1994; Daire et al., JBC, 2009), and the presence of kif5b at the mitochondrion, ERES, ERGIC and endolysosome is hardly detectable. In our case, we'd like to argue that the small amount of Dopey1 and Mon2 is probably sufficient to drive the motility of the ERES, ERGIC and endolysosome despite their majority localization at the Golgi.

Reference:

Marks DL, Larkin JM, McNiven MA. (1994) Association of kinesin with the Golgi apparatus in rat hepatocytes. *J Cell Sci.* 107 (Pt 9):2417-2426.

Daire V, Giustiniani J, Leroy-Gori I, Quesnoit M, Drevensek S, Dimitrov A, Perez F, Poüs C (2009) Kinesin-1 regulates microtubule dynamics via a c-Jun N-terminal kinase-dependent mechanism. *J Biol Chem.* 284(46):31992-2001.

Moreover, it is surprising not to see the various mutations defined in the early part of the manuscript not incorporated at all into the later functional analysis – for example, determining the effect of the

LRRK/4A and W/A mutations on the trafficking/positioning phenotypes.

Reply:

Please see our replies to point 2 and 3 (of reviewer #2).

Specific Comments:

1. I am not convinced that the data presented in Figure 1 adequately support the very explicitly stated stoichiometry of the complex. Whilst the data are indeed consistent with at least a dimeric Mon2 that associates with 2x Dopey1, a higher order assembly also seems possible. Unless, more direct evidence can be provided (perhaps using purified recombinant proteins and appropriate biophysical analysis), the authors should tone down this conclusion. Regardless, it would seem relatively straightforward to formally demonstrate the direct domain-domain interactions within the complex and with KLC that are inferred from the co-IP experiments.

Reply:

We have performed the single molecule photo-bleaching experiment to directly measure the stoichiometry of cellular overexpressed GFP-Mon2, GFP-Dopey1 and GFP-Dopey1/DMyC-Mon2 complexes (Fig. 1g and Supplementary Fig. 11,m in revamped manuscript). We found that $\geq 90\%$ of GFP-Mon2 and GFP-Dopey1 objects are in homodimer and monomer status. Co-overexpression of DMyC-Mon2 can greatly increase the dimeric GFP-Dopey1 by more than 10-fold. Our new data therefore demonstrate the stoichiometry that we previously claimed.

We also provide further data to conclude that the following three interactions are direct: MEC and MEC, MEC and Dopey1, and DEN and TPR interactions. Please see our reply to point 5 of reviewer #1's comment.

2. The authors implicate a tryptophan residue in the DEN domain as important for KLC2 binding. The various western blot data supporting this are convincing, but the authors mechanistically connect this previously reported WD motifs found in some other kinesin-1 adaptors, despite some sequence differences. Moreover, the WA motif appears to be found within a predicted alpha helical region (LZ?) of Dopey1, whereas validated WD motifs have so far been found in predicted unstructured regions of proteins. The authors suggest in their schematic that is region is a potential leucine zipper. They might consider the extensive literature on JIP3/4-KLC binding which appears to occur via a LZ-TPR interaction in their narrative. Regardless, the authors should incorporate this mutation into the functional analysis of peroxisome relocalization, organelle positioning and Golgi-trafficking to demonstrate the importance of the link to kinesin-1 for its function.

Reply:

The "leucine zipper-like" region was from the annotation in NCBI Conserved Domains (Accession No.: COG5221). This reviewer's comment prompted us to conduct further bioinformatics investigation of these regions. We found that Dopey1 should not have coiled-coils and leucine zippers from online prediction tools such as 2Zip (<http://www.lirmm.fr/2zip/>) and coils (https://embnet.vital-it.ch/software/COILS_form.html). We therefore think that "leucine zipper-like" region predicted by NCBI Conserved Domains is incorrect (or at least misleading). We have removed these annotations in the revamped version.

As suggested by this reviewer, we constructed GFP-DEN-FRB with W34A mutation for peroxisomal translocation assay. When co-expressed with PEX3-mRFP-FKBP, GFP-DEN-FRB (W34A) is unable to drive the centrifugal movement of peroxisomes in the presence of rapamycin (Supplementary Fig. 5l,m in the revamped manuscript). Therefore, the new result is consistent with our finding that the WD motif of DEN is required for the interaction with kinesin-1 motor.

3. Similarly, the authors should determine the effect of lipid binding mutations in their positioning/trafficking functional assays.

Reply:

We constructed GFP-Dopey1 full length with LKRL-4A mutation and examined the cellular distribution of lysosomes upon its overexpression. However, we didn't observe any change in normalized gyradius of lysosomes, suggesting that overexpressed GFP-Dopey1 with LKRL-4A mutation might not behave as a dominant negative mutant. A possible explanation is that LKRL-4A cannot compete with

endogenous wild type Dopey1 for associating with Mon2 due to its inability to bind to the membrane. A better approach would be to directly mutate LKRL to 4A in the native genomic copy of Dopey1 gene. Take note that, since DEN region is still intact, Mon2-binding Dopey1(LKRL-4A) should be able to recruit kinesin-1 motor. Therefore, there might not be a strong disruption in the organelle positioning and membrane carrier movement. Due to the time constraint, we propose to reserve this experiment for our future project.

4. Despite the quantification, I would like to see representative fields of cells, not just single cells when describing the organelle positioning phenotypes as this can vary considerably from cell to cell in a single experiment and the extent of these changes is not readily apparent from simply considering the gyradius measurement provided. In addition, I cant see a description of this quantitative approach in the methods as the main text indicates.

Reply:

As suggested, we have prepared organelle positioning images showing fields of cells in Supplementary Fig. 5f (revamped version). Cells look a bit smaller but we hope that the images are still clear enough to demonstrate that the change of organelle positioning occurs in the majority of cells.

The description of normalized gyradius was previously in the "Image analysis" of Supplementary Information. It has been moved to the Methods in the revamped version.

5. The authors use the SUK4 antibody to detect Kif5B. They should confirm the specificity of this antibody in their IF experiments using siRNA knockdown.

Reply:

The Golgi localization of kinesin-1 by using SUK4 antibody has been previously reported (Marks et al., JCS, 1994; Daire et al, JBC, 2009). To further verify the specificity of this antibody, we have designed a shRNA against kif5b, as suggested by this reviewer. When the kif5b shRNA, but not the control shRNA, was expressed by the lentivirus transduction, the Golgi staining of SUK4 greatly reduced (as shown below), demonstrating the specificity of SUK4 antibody. The result is not included in our manuscript.

Reference:

Marks DL, Larkin JM, McNiven MA. (1994) Association of kinesin with the Golgi apparatus in rat hepatocytes. *J Cell Sci.* 107 (Pt 9):2417-2426.

Daire V, Giustiniani J, Leroy-Gori I, Quesnoit M, Drevensek S, Dimitrov A, Perez F, Poüs C (2009) Kinesin-1 regulates microtubule dynamics via a c-Jun N-terminal kinase-dependent mechanism. *J Biol Chem.* 284(46):31992-2001.

6. It is not clear why the proteomic analysis that informs these studies is not described in the supplementary data.

Reply:

Methods of large scale IP and proteomic analysis have been added in the Supplementary Information.

7. Figure1A - please show directly comparable regions of the reprobed upper and lower panels i.e. the region of the Mon2 blot that corresponds to the location of the Dopey2 band the Dopey2 reprobe.

Reply:

We have modified Fig. 1a (the revamped version) to include the corresponding blot region before Dopey2 re-probing (the last row).

8. What effect does Mon2 knockdown have on Dopey2 Golgi-localisation. Is this consistent with the binding data in figure 1?

Reply:

We have performed the experiment and found that Mon2 knockdown does not affect the Golgi localization of Dopey2. The new data has been added into the main text and presented as Supplementary Fig. 2f (the revamped version). The result is consistent with binding data shown in Fig. 1, which demonstrates that Dopey2 does not interact with Mon2.

9. In general, molecular weight maker annotations should be provided for the western blots.

Reply:

In the revamped version, we have labelled molecular weight markers (in kDa) in all Coomassie gels and Western blots.

10. The authors state that low pH buffer 'activates kinesin-1' based on work by Heuser. This overstates current knowledge – for still unclear reasons, this effect is quite specific to lysosomes and late endosomes and may or may not be caused by a direct activation of kinesin-1 or could regulate upstream, lysosome specific pathways.

Reply:

Thanks for pointing this out. We have modified our text accordingly to "Low cytoplasmic pH is known to cause the kinesin-1-dependent peripheral distribution of lysosomes, of which the molecular mechanism is still unclear. Using Ringer's acetate solution (pH6.5)(Supplementary Fig. 5e), we demonstrated that Dopey1 and Mon2 are required for acidification-induced centrifugal positioning of lysosomes".

11. Figure 3 – the effect of PAO on Dopey1 Golgi localisation should be quantified. There also appears to be some reduction in Golgi 245 staining that may confound the interpretation.

Reply:

We have quantified the kinetics of Golgi intensity changes of Dopey1, Dopey2, Mon2, GFP-Dopey1, GFP-DEC, GFP-tdDEC and mCherry-P4M during PAO treatment. The new data are presented in Fig. 3b,d,f and Supplementary Fig. 3d,f,n,o (the revamped version). As this reviewer noticed, there is a slight, ~ 16%, reduction of Golgi intensity of Golgin245 after 10 min of PAO treatment (data not shown), probably due to an unknown specific effect of PAO on the Golgi association of Golgin245. However, it should not confound our interpretation that Dopey1 and DEC are sensitive to PAO, since 1) the Golgi intensities of Dopey2 and Mon2 remained constants and 2) intensities of Dopey1 and DEC dropped far more than 16%.

12. For the liposome co-sed experiments, how do the authors know that all liposomes formed and pelleted with similar efficiency, which would be important for their interpretation of binding specificity?

Reply:

The protocol we used is standard according to our knowledge. We also investigated the concern raised by this reviewer by performing the below control experiment.

The base-liposome (69% PC, 14% PE and 17% PS) (mass percentage or m/m, same for the rest) was first prepared in chloroform. To prepare liposome containing phosphoinositide or PA-derivative, 10% PC of the base-liposome was substituted by PI, PA or PI4P together with 0.01% Dil (Invitrogen). A thin lipid film was resulted after chloroform was removed in the vacuum. The lipid film was rehydrated in a buffer (120 mM NaCl, 40 mM HEPES, pH7.3) to obtain a final concentration of 1 mg/ml total lipids. Giant liposomes were subsequently generated after bath sonication. Equal amount of liposomes containing PI, PI4P or PA were first mixed with HEK293T cell lysate at room temperature for 1 h. Next, they were pelleted in a table top centrifuge at 17,000g for 10 min. After the pellet was washed and dried in vacuum, it was dissolved in DMSO and the fluorescence was monitored by Tecan Infinite M200 pro

at 25 °C with the excitation at 530 nm and emission at 570 nm. The Dil fluorescence intensity measured (arbitrary unit) for pellets from base, PI, PA and PI4P liposomes are: 113 ± 2 , 116 ± 8 , 113 ± 6 and 112 ± 4 respectively (mean \pm s.d.; n=3 experiments). Our result (shown below) indicated that the same amount of Dil was recovered from different preparations of liposomes, strongly supporting that different types of liposomes form and pellet with similar efficiency.

Reviewer #3 (Remarks to the Author):

The authors present a very interesting and comprehensive analysis of a novel kinesin-1 adaptor complex, the Dopey1-Mon2 complex. What is particularly engaging is the thorough characterisation of the lipid-mediated recruitment (which is very novel in the kinesin field) and the highly quantitative characterisation of organelle distribution changes in response to perturbations. This work will be of broad interest to the cell biology community and I recommend publication with minor revisions. My comments are outlined below in rough order of priority.

1) In general there is not enough description of how analyses were performed (e.g. golgi localisation quantification), and in particular which statistical tests were performed to generate the p values. A description of analysis methodology is needed and which tests were used in the figure legends. I realise that single cell analysis has been carried out for the most part, but there is also no clear indication of how many biological replicates were done for each experiment (presumably at least three?). In general the manuscript comes across as comprehensive and well considered so I have concluded that this omission was by accident rather than design - although it is obviously difficult to review statistical rigor without these details. These omissions are throughout the manuscript in all figures.

Reply:

- (1) Image analysis was described in the Supplementary Information in the previous version. We now moved it to the Methods in the revamped version. As suggested by this reviewer, we also provided more details on our protocols in Methods.
- (2) All *P*-values are from unpaired and two-tailed *t* test. It is now stated in every relevant Figure legend together with the range of *P* values in number of asterisks (*).
- (3) All experiments have at least 3 biological replicates. However, depending on experiments, results are presented in one of the following 3 ways: (I) as the average of 3 biological replicates, (II) as a cumulative combination using data from 3 biological replicates (e.g., particle analysis) and (III) by using one of the representative data. We have indicated this information clearly in the figure legends in the revamped version.

2) In the methods it is apparent that both HeLa cells and HEK cells are used, but it is unclear which cell lines are used for which experiment when reading the manuscript or referring to the figure legends.

Reply:

We previously only stated in Methods (cell culture and transfection) that "All cells used in imaging are HeLa cells. All cells used in IPs, pull-downs and immuno-blots are 293T cells unless specified otherwise". In the revamped version, we have incorporated this information in all figure legends.

3) In general the authors overstate their confidence in the stoichiometry of the complex and the 'directness' of interactions given the experiments they have done. Although I agree that Mon2 and Dopey1 likely multimerise, statements about dimerisation and heterotetramerisation - particularly in the first part of the manuscript (lines 58 to 82, including the subtitle) - can't really be justified. Later on with the artificial dimerisation it does suggest this for Dopey, but I would not conflate this with the earlier results. Similarly, in line 211: replace "directly interacts" with complexes. Avoid the speculation of bivalent interactions at line 218 (it's somewhat under debate how many WD motifs are needed to activate kinesin through KLC2).

Reply:

(1) Stoichiometry:

We have adopted a single molecule photo-bleaching method to directly quantify the number of GFP-tagged molecules. We now have evidence to demonstrate that Mon2 alone is a homodimer while Dopey1 alone is a monomer. Please see our reply to point 1 of reviewer #1's comment.

(2) Directness of interactions:

We have demonstrated most of them by performing interaction or pull-down assays using in vitro translated or purified recombinant proteins. Please see our reply to point 5 of reviewer #1's comment.

(3) Bivalent interaction:

As suggested, we have deleted the sentence "The enhancement makes sense since Mon2-homodimerized Dopey1 probably displays bivalent interaction with homodimeric KLC2".

4) The author's maintain that the DEC domain is not conserved between Dopey1 and Dopey2 (line 106), however a quick look at the alignment between these two shows that the C-terminal sequences are broadly very similar. In fact the LKRL motif and flanking residues of Dopey1 are almost identical in Dopey2, but the motif carries a change to LKRQ. I would say the striking difference between Dopey1 and Dopey2 is even more interesting in this light, rather than trying to say the C-terminus of Dopey1 is entirely unique (line 107). Please adjust the way this section is phrased to better reflect the sequence identity between Dopey1 and Dopey2.

Reply:

As this reviewer pointed out, we agree that the DEC corresponding region of Dopey2 is very similar to DEC. We have prepared a multiple alignment to illustrate this point (Fig. 2h in the revamped version). Our text has been modified to reflect the similar sequence between Dopey1 and 2 in DEC corresponding region. The statement that DEC is unique to Dopey1 has been removed in the new version of our manuscript.

5) In general I find the author's use of "DEC", "DEN", "MEC" etc to be quite confusing and a little meaningless. There are no such domains, just protein fragments. In the case of "DEC" it seems that it is not the DEC domain that is necessary for targeting to the Golgi, but the LKRL motif. Similarly it is not the DEN domain that binds to KLC2, but the WD motif. We can agree to disagree, I only ask that the author's consider whether it really adds to the clarity of the paper.

Reply:

We take this reviewer's advices and modified our manuscript accordingly. Now we refer "DEC", "DEN" and "MEC" as "region" instead of "domain".

We think that the LKRL motif is necessary but insufficient for Dopey1's targeting to the Golgi since, in Fig. 2f (revamped version), the fragment encompassing residues 2101-2408, which contains intact LKRL motif, did not show Golgi localization. Thus, in addition to LKRL motif, other sequence within DEC are also required. Similarly, in Fig. 5b (revamped version), other sequences within DEN are required for the interaction with KLC2, in addition to W34. Therefore, LKRL and WD-motif are essential for the interaction but other regions also contribute to the binding.

6) Supplemental Figure 2C,D (line 88 in manuscript), the siRNA knockdown and stability isn't quantified so it is difficult to gauge how reproducible the knockdown was.

Reply:

We have quantified 3 independent sets siRNA knockdown Western blots and the result is shown in Supplementary Fig. 2d (revamped version).

7) Lines 100-101: you mention a highly conserved LKRL motif from analysing sequences, could you include a cross species alignment to verify this for the reader?

Reply:

As suggested by this reviewer, a multiple alignment has been prepared as Fig. 2h (revamped version).

8) Line 105: Please show images that reflect the cytosolic distribution of Dopey2 C-terminal fragments as claimed.

Reply:

As suggested by this reviewer, we have added the indicated Dopey2 images in Supplementary Fig. 2h (revamped version).

9) Please quantify Figure 3A-C in a similar manner to Figure 2D/E, but showing the changes over time. Similarly Supplemental Figure 3I.

Reply:

As suggested by this reviewer, we have quantified the kinetics of Golgi intensity changes of Dopey1, Dopey2, Mon2, GFP-Dopey1, GFP-DEC, GFP-tdDEC and mCherry-P4M during PAO treatment. The new data are presented in Fig. 3b,d,f and Supplementary Fig. 3d,f,n,o (the revamped version).

10) Has Supplemental Figure 3J,K been quantified? The P14P signal looks less in the Mon2 siRNA condition.

Reply:

As suggested by this reviewer, they are now quantified and shown as Supplementary Fig. 3q,s (revamped version). The change of PI4P is not significant (P value >0.05). More representative images have been used in Supplementary Fig. 3p (previously Supplementary Fig. 3J).

11) Is the multiple vesicular organelle (e.g. EE, LE, lysosome etc etc) localisation of FP-tagged Mon2 and Dopey1 (Fig 5A and Sup Fig 5A) an overexpression artifact - do you see this localisation using endogenous antibodies?

Reply:

Our Mon2 polyclonal antibodies showed many fine intracellular puncta during immunofluorescence staining (Supplementary Fig. 6b of the revamped version). Majority of these puncta are likely specific as these signals disappear upon Mon2 knockdown or in the presence of its corresponding antigen. Some colocalization of Mon2 with Tf receptor and Lamp1 can be observed, demonstrating the endogenous presence of Mon2 at the endolysosome. Our Dopey1 polyclonal antibody produces an immuno-staining background that is too noisy to be useful for its possible endolysosomal localization study.

12) Line 171: Did the ACC1-dimer also dissociate?

Reply:

The result is now shown in Supplementary Fig. 4c (revamped version). As expected and in contrast to Dopey1 (Supplementary Fig. 4b), ACC1-dimer of Dopey1 does not dissociate from the Golgi under 1-butanol treatment due to its increased binding avidity toward PI4P. Therefore, ACC1-dimer can associate with the Golgi membrane independent of Mon2.

13) Supplemental Figure 3 has a leftover placeholder box in the bottom right hand corner.

Reply:

We have deleted the unwanted box, which was accidentally left there.

14) Make in clear in the text (line 192) that Figure 4A is endogenous and SF4A is overexpressed.

Reply:

We believe that these have been indicated in the previous version of manuscript. Those sentences were copied below with endogenous and overexpression underlines.

“The interaction between Dopey1 and Kif5b was confirmed by subsequent endogenous co-IPs. We found that, in addition to Mon2, Dopey1, but not Dopey2, specifically co-IPed Kif5b, but not Kif20a and Kif18a (Fig. 4a).”

“By overexpression, we found that Dopey1 co-IPed KLC2 and co-expression of KLC2 was required for Dopey1 to co-IP Kif5b (Supplementary Fig. 4a)”.

15) Is there a graph for Figure 4E? How many repeats were done.

Reply:

Fig. 4E has become Fig. 5e in the revamped version. There biological replicates have been performed and gel blots are shown below. We didn't generate a graph or plot. Instead, the quantification values are directly indicated above corresponding bands. The disproportionately large amount of KLC2-pull-down in the presence of overexpressed Mon2 were observed in all three experiments.

Reviewers' comments:

Reviewer #1 (Remarks to the Author):

The authors have only partially addressed the concerns raised by this reviewer.

1. As suggested, the authors have performed the KD of PI4KIIIbeta (though they have not used very specific inhibitors that are now commercially available), and they still use only PAO to address the issue of PI4P dependence of the localization of Dopey to non-Golgi areas. However, there are major problems with Fig. 3 (possibly due to the small size and poor quality of the images): in Fig. 3c the merge panels should show some kind of yellow pattern (resulting from green Dopey colocalizing with red Golgin245) under control conditions (0 PAO) but only green is visible (?) and after PAO-induced dissociation of Dopey one should get just a red (Golgin245) pattern: there is apparently nothing left (neither green nor red) after PAO, possibly because PAO induces fragmentation of the Golgi complex (?). Fig 3e: the merge images just show green marker (?). Fig 3g: the overall quality is so low that it is impossible to get any information.

2. The inability to reproduce alcohol-induced phenotypes by KD of PLDs casts doubt on the specificity of these phenotypes. A way to circumvent this and to acutely inhibit PLD (as the authors suggest that the acute, but not the chronic, inhibition of PLD could be effective) would be to use more specific PLD inhibitors that are commercially available (FIPI, VU0155056, VU0359595, VU0285655-1, VU0155069). In the absence of these important controls and in the absence of a clear demonstration of the direct binding of PA to Mon2, all the data on PA requirements remain unconvincing. In spite of recognizing that there are no data showing a direct interaction of PA with Mon2, the authors state repetitively in the revised manuscript that Mon2 binds PA:

"Further truncation of Mon2(1-1035) abolished its PA-binding (Fig. 4e), suggesting that the fragment comprising residue 1-1035, including conserved DCB, HUS and HDS1-2 regions, is the smallest PA-binding region. However, Mon2(1-1035) did not localize to the Golgi in vivo (Fig. 4a,b), in contrast to Mon2(1-1200), demonstrating that PA-binding is required but is probably insufficient for the Golgi localization of Mon2. The elucidation of residues that engage PA and additional potential factor(s) for the Golgi localization awaits further investigation. In summary, our data so far demonstrate that Dopey1-Mon2 complex probably engages two types of Golgi resident lipids, PA and PI4P, for its Golgi targeting."

Legend Fig. 4 "The Golgi localization of Mon2 requires its binding to P 942 A. "

Scheme in Fig 9b:

The suggestion of this reviewer is to convincingly prove the requirement for and the interaction with PA. In the absence of convincing data the alternative is to remove all of the data set concerning the involvement of PLDs/PA.

3. This is a residual minor concern: in principle I could agree with the authors explanation. However, as the specific signal of GFP-Dopey1(1900-2456) is so low, it would be preferable to have it on the same blot together with the negative control (as the authors did for the GST-DEC region in panel n)

4. The interaction of Mon2 with PA: see above point 2.

5. The authors have satisfactorily addressed the issue of the direct interactions between MEC and MEC, MEC and Dopey1, and DEN and TPR. What remains unexplained is the disproportionate amount of KLC2 that is pulled down by PI4P liposomes in the presence of Dopey and Mon2. In fact, the huge amount of KLC2 that is pulled down (in large excess compared to the amount of Dopey), is not consistent with the 1:1 stoichiometry (Dopey:KLC2) depicted in Fig. 1 and indicates either that one Dopey molecule can bind multiple KLC2 molecules, or that there is an unknown "multiplying" factor that is pulled down by PI4P liposomes in the presence of Mon2 and Dopey1 that engages multiple KLC2 molecules per Dopey molecule.

6. The authors have addressed my concerns (although they have used the non-specific kinase inhibitor PAO and not more specific tools).

Reviewer #2 (Remarks to the Author):

The authors have performed a significant amount of new work in this revised manuscript that has enhanced the robustness of the conclusions. In particular, the detection of endogenous Mon2 on other organelles makes their arguments more coherent - I would consider moving this from supplementary into main figures.

I would have still liked to have seen rescue of siRNA phenotypes with lipid/kinesin binding mutant versions of proteins and the response doesn't address why this isn't possible. This wouldn't require genome editing as stated and is more likely to produce useful data than a potential dominant negative.

Nonetheless, there is a huge amount of work here, the conclusions potentially very important in understanding organelle dynamics and positioning, and their case compellingly made, and so I don't object to the publication of the manuscript in its current form.

Reviewer #3 (Remarks to the Author):

The author's have answered all my previous comments - this is a detailed and informative piece of work and I look forward to its publication.

Point-by-point reply to reviewers' comments

Reviewer #1 (Remarks to the Author):

The authors have only partially addressed the concerns raised by this reviewer.

1. As suggested, the authors have performed the KD of PI4KIIIbeta (though they have not used very specific inhibitors that are now commercially available), and they still use only PAO to address the issue of PI4P dependence of the localization of Dopey to non-Golgi areas. However, there are major problems with Fig. 3 (possibly due to the small size and poor quality of the images): in Fig. 3c the merge panels should show some kind of yellow pattern (resulting from green Dopey colocalizing with red Golgin245) under control conditions (0 PAO) but only green is visible (?) and after PAO-induced dissociation of Dopey one should get just a red (Golgin245) pattern: there is apparently nothing left (neither green nor red) after PAO, possibly because PAO induces fragmentation of the Golgi complex (?). Fig 3e: the merge images just show green marker (?). Fig 3g: the overall quality is so low that it is impossible to get any information.

Reply:

We apologize for the poor quality of Figure 3c,e,g. We have rescaled the intensity of all images in Figure 3c,e,g,h. Dotted lines marking cell contours were removed in Figure 3g as they seemed to interfere with image viewing. We hope these revamped images appear fine now.

2. The inability to reproduce alcohol-induced phenotypes by KD of PLDs casts doubt on the specificity of these phenotypes. A way to circumvent this and to acutely inhibit PLD (as the authors suggest that the acute, but not the chronic, inhibition of PLD could be effective) would be to use more specific PLD inhibitors that are commercially available (FIPI, VU0155056, VU0359595, VU0285655-1, VU0155069). In the absence of these important controls and in the absence of a clear demonstration of the direct binding of PA to Mon2, all the data on PA requirements remain unconvincing. In spite of recognizing that there are no data showing a direct interaction of PA with Mon2, the authors state repetitively in the revised manuscript that Mon2 binds PA:

“Further truncation of Mon2(1-1035) abolished its PA-binding (Fig. 4e), suggesting that the fragment comprising residue 1-1035, including conserved DCB, HUS and HDS1-2 regions, is the smallest PA-binding region. However, Mon2(1-1035) did not localize to the Golgi in vivo (Fig. 4a,b), in contrast to Mon2(1-1200), demonstrating that PA-binding is required but is probably insufficient for the Golgi localization of Mon2. The elucidation of residues that engage PA and additional potential factor(s) for the Golgi localization awaits further investigation. In summary, our data so far demonstrate that Dopey1-Mon2 complex probably engages two types of Golgi resident lipids, PA and PI4P, for its Golgi targeting.”

Legend Fig. 4 “The Golgi localization of Mon2 requires its binding to P 942 A. ”

Scheme in Fig 9b:

The suggestion of this reviewer is to convincingly prove the requirement for and the interaction with PA. In the absence of convincing data the alternative is to remove all of the data set

concerning the involvement of PLDs/PA.

Reply:

Multiple metabolic pathways from phosphatidylcholine (PC), diacylglycerol (DAG) and lysophosphatidic acid (LPA) can lead to the production of PA. As suggested by this reviewer, we tested CAY10593 (VU0155069)(a PLD1 inhibitor), CAY10594(a PLD2 inhibitor), FIPI (a PLD1/2 inhibitor), CI-976 (a lysophosphatidic acid acyl transferase inhibitor) and R59949 (a diacylglycerol kinase inhibitor). The result is shown below as a bar graph. CAY10593 and CAY10594 did not significantly reduce Golgi intensity of Mon2 (endogenous); on the other hand, FIPI significantly increased the Golgi intensity of Mon2. The result is unexpected but it is consistent with previous findings that inhibition of PLD1/2 by inhibitors or knockdown does not reduce the cellular level of PA (an increase has been reported instead) (Morita et al., 2009; Antonescu et al., 2010).

It has been reported that the majority of cellular PA can be depleted by R59949 (Antonescu et al., 2010; Giridharan et al., 2013). We observed >50% reduction of the Golgi-localized Mon2 after R59949 treatment. Golgi-localized Mon2 only marginally reduced under CI-976 treatment.

Figure 1 HeLa cells were treated with 1 μ M indicated inhibitors (dissolved in DMSO) or equal volume of DMSO diluted in DMEM + 10% FBS for 1 h at 37°C before PFA fixation and immunofluorescence labeling of endogenous Mon2 and a Golgi marker.

Due to the potential pleiotropic effect of primary alcohol, we adopt this reviewer's advice by removing data acquired by using the primary alcohol. In Figure 4c,d and Supplementary Figure 4b-e (revamped version), figure panels using primary alcohol were replaced by those performed using R59949. Corresponding quantifications were also included. Supplementary Figure 7 f-j, which shows particle analysis under primary alcohol treatment, were removed.

In combination with our PA-liposome binding assay, our diacylglycerol kinase inhibitor data made it more convincing that Mon2 might interact with PA, although we were unable to acquire purified or in vitro translated Mon2(1-1035) to test if their interaction is direct or indirect. Although we stated in our manuscript that Mon2 binds to PA, we did not mean that they directly interact. To make it clearer about our uncertainty of direct or indirect interaction, we have modified our manuscript to explicitly state it, as shown below.

(Regarding the interaction) "Further truncation of Mon2(1-1035) abolished its PA-binding (Fig. 4f), suggesting that the fragment comprising residue 1-1035, including conserved DCB, HUS and HDS1-2 regions, is the smallest region that either directly or indirectly interacts with PA. However, Mon2(1-1035) did not localize to the Golgi in vivo (Fig. 4a,b), in contrast to 1-1200, demonstrating that PA-binding is required but is probably insufficient for the Golgi localization of Mon2. Resolving direct or indirect PA-binding and identifying additional factor(s) for Mon2's Golgi localization await further investigation."

(Regarding our model, in the legend of Fig. 9b) “Its N-terminal region interacts with PA (directly or indirectly).”

References:

Antonescu CN, Danuser G, Schmid SL. (2010) Phosphatidic acid plays a regulatory role in clathrin-mediated endocytosis. *Mol. Biol. Cell*, 21(16):2944-52.

Giridharan SS, Cai B, Vitale N, Naslavsky N, Caplan S. (2013) Cooperation of MICAL-L1, syndapin2, and phosphatidic acid in tubular recycling endosome biogenesis. *Mol. Biol. Cell*, 24(11):1776-90, S1-15.

Morita SY, Ueda K, Kitagawa S. (2009) Enzymatic measurement of phosphatidic acid in cultured cells. *J. Lipid Res.*, 50(9):1945-52.

3. This is a residual minor concern: in principle I could agree with the authors explanation. However, as the specific signal of GFP-Dopey1(1900-2456) is so low, it would be preferable to have it on the same blot together with the negative control (as the authors did for the GST-DEC region in panel n)

Reply:

Figure 3l,m of the previous version manuscript showed the PI4P-liposome pull-down of GFP-Dopey1(1900-2456) wild type and LKRL-4A respectively. They were cropped from the same Western blot PVDF membrane. As this reviewer suggested, both pull-down experiments are now shown in the same uncropped blot image of Figure 3l in the revamped version.

4. The interaction of Mon2 with PA: see above point 2.

Reply:

Please see our reply to point 2 of reviewer #1's comments.

5. The authors have satisfactorily addressed the issue of the direct interactions between MEC and MEC, MEC and Dopey1, and DEN and TPR. What remains unexplained is the disproportionate amount of KLC2 that is pulled down by PI4P liposomes in the presence of Dopey and Mon2. In fact, the huge amount of KLC2 that is pulled down (in large excess compared to the amount of Dopey), is not consistent with the 1:1 stoichiometry (Dopey:KLC2) depicted in Fig. 1 and indicates either that one Dopey molecule can bind multiple KLC2 molecules, or that there is an unknown "multiplying" factor that is pulled down by PI4P liposomes in the presence of Mon2 and Dopey1 that engages multiple KLC2 molecules per Dopey molecule.

Reply:

The disproportional increase in the recruited Flag-KLC2 has been observed in three independent experiments (see the below image which was copied from the previous rebuttal letter in our reply to point 15 of reviewer #3's comment). It is due to the enhanced binding efficiency between Dopey1 and KLC2 in the presence of Mon2.

Dopey1 alone is sufficient to bind to KLC2 (as evidenced by the direct interaction between DEN and TPR) to form a complex with 1:1 stoichiometry on the PI4P-liposome. However, the interaction is likely inefficient. When both Dopey1 and KLC2 are abundant, the percentage of Dopey1 that engages KLC2 is low (Figure 5e; the lane with positive PI4P, GFP-Dopey1 and Flag-KLC2). Since the dimer-dimer interaction is stronger than monomer-monomer one, Mon2-dimerized Dopey1 might interact with KLC2 much more efficiently, resulting in the disproportional recruitment of KLC2 (Figure 5e; the last lane of pull-down blot).

6. The authors have addressed my concerns (although they have used the non-specific kinase inhibitor PAO and not more specific tools).

Reply:

Thanks.

Reviewer #2 (Remarks to the Author):

The authors have performed a significant amount of new work in this revised manuscript that has enhanced the robustness of the conclusions. In particular, the detection of endogenous Mon2 on other organelles makes their arguments more coherent - I would consider moving this from supplementary into main figures.

I would have still liked to have seen rescue of siRNA phenotypes with lipid/kinesin binding mutant versions of proteins and the response doesn't address why this isn't possible. This wouldn't require genome editing as stated and is more likely to produce useful data than a potential dominant negative.

Nonetheless, there is a huge amount of work here, the conclusions potentially very important in understanding organelle dynamics and positioning, and their case compellingly made, and so I don't object to the publication of the manuscript in its current form.

Reply:

As suggested by this reviewer, the localization of endogenous Mon2 to the endolysosomal compartment is included in Figure 6b (revamped version).

We were unsuccessful in cloning siRNA-resistant Dopey1 expression construct that contains either LKRL-4A or W34A mutation. An alternative approach was adopted by using Dopey1-ACC1 to test the effect of LKRL-4A or W34A mutation. Dopey1-ACC1 is resistant to Dopey1 siRNA since siRNA target region has been replaced by ACC1. Furthermore, it is much easier to clone Dopey1-ACC1 with LKRL-4A or W34A mutation since the coding sequence of Dopey1-ACC1 (3.1 kb) is much shorter than that of Dopey1 (7.4 kb). When HeLa cells were subjected to Dopey1 knockdown, lysosomes appeared aggregated at the perinuclear region, as expected. We found that Dopey1-ACC1 wild type, but not W34A or LKRL-4A mutant, can rescue the phenotype by dispersing lysosomes, therefore highlighting the importance of W34 and LKRL. These new results are included as Supplementary Figure 6n,o (revamped version).

Reviewer #3 (Remarks to the Author):

The author's have answered all my previous comments - this is a detailed and informative piece of work and I look forward to its publication.

Reply:

Many thanks.

REVIEWERS' COMMENTS:

Reviewer #1 (Remarks to the Author):

The authors have satisfactorily addressed the concerns of this reviewer.

REVIEWERS' COMMENTS:

Reviewer #1 (Remarks to the Author):

The authors have satisfactorily addressed the concerns of this reviewer.

Reply:

Thanks. We appreciate your review comments very much.